# Generalized stability landscape of the Atlantic Meridional Overturning Circulation

Matteo Willeit[1,*] and Andrey Ganopolski[1,*]

[1]Potsdam Institute for Climate Impact Research (PIK), Member of the Leibniz Association, P.O. Box 601203, D-14412 Potsdam Germany
[*]These authors contributed equally to this work.

**Correspondence:** Matteo Willeit (willeit@pik-potsdam.de)

**Abstract.**

The Atlantic Meridional Overturning Circulation (AMOC) plays a crucial role in shaping climate conditions over the North Atlantic region and beyond and its future stability is a matter of concern. While the AMOC stability to surface freshwater forcing (FWF) has been thoroughly investigated, its equilibrium response to changing $CO_2$ remains largely unexplored, precluding a comprehensive understanding of its stability under global warming. Here we use an Earth system model to explore the stability of the AMOC to combined changes in FWF in the North Atlantic and atmospheric $CO_2$ concentrations between 180 and 560 ppm. We find four different AMOC states associated with qualitatively different convection patterns. Apart from an *Off* AMOC state with no North Atlantic deep water formation and a *Modern*-like AMOC with deep water forming in the Labrador and Nordic Seas as observed at present, we find a *Weak* AMOC state with convection occurring south of 55°N and a *Strong* AMOC state characterized by deep water formation extending into the Arctic. The *Off* and *Weak* states are stable for the entire range of $CO_2$, but only for positive FWF. The *Modern* state is stable under higher than preindustrial $CO_2$ for a range of positive FWF and for lower $CO_2$ only for negative FWF. Finally, the *Strong* state is stable only for $CO_2$ above 280 ppm and FWF<0.1 Sv. Generally, the strength of the AMOC increases with increasing $CO_2$ and decreases with increasing FWF. Our AMOC stability landscape helps to explain AMOC instability in colder climates and, although it is not directly applicable to the fundamentally transient AMOC response to global warming on a centennial time scale, it can provide useful information about the possible long-term fate of the AMOC. For instance, while under preindustrial conditions the AMOC is monostable in the model, the *Off* state also becomes stable for $CO_2$ concentrations above ∼400 ppm, suggesting that an AMOC shutdown in a warmer climate might be irreversible.

## 1 Introduction

The Atlantic Meridional Overturning Circulation (AMOC) is a critical component of the global climate system and has been extensively studied due to the large climate implications that a change in this circulation would cause in the North Atlantic region, particularly over Europe (Jackson et al., 2015). There is a concern that the AMOC could weaken substantially or even shut down in the future under global warming (e.g. Manabe and Stouffer, 1993; Weaver et al., 2012; Weijer et al., 2020; Bellomo et al., 2021), and that this could be possibly irreversible due to the existence of multiple equilibrium states (Manabe

and Stouffer, 1988; Rahmstorf et al., 2005; Mecking et al., 2016; Jackson and Wood, 2018) in accordance with the seminal work of Stommel (1961), who suggested the presence of multiple stable AMOC states due to the positive salt advection feedback.

Stocker and Wright (1991) and Rahmstorf (1995) pioneered the use of surface freshwater forcing (FWF) experiments to analyze the stability of the AMOC and showed a hysteresis behavior in ocean models. Since then, models of different complexity have found that the AMOC shows a hysteresis behavior to FWF that is associated with multiple stable states (Ganopolski and Rahmstorf, 2001; Gregory et al., 2003; Rahmstorf et al., 2005; Lenton et al., 2009; Hofmann and Rahmstorf, 2009; Hawkins et al., 2011; Hu et al., 2012; Ando and Oka, 2021; van Westen and Dijkstra, 2023), although there is no consensus as to whether the AMOC is in a monostable or a bistable regime under present climate conditions (e.g. Weijer et al., 2019; Liu et al., 2017). While most of these hysteresis experiments have been performed under pre-industrial or present-day conditions, some have considered the dependence on background climate by exploring the hysteresis behavior also for the last glacial maximum (Ganopolski and Rahmstorf, 2001; Schmittner et al., 2002; Prange et al., 2002; Weber and Drijthout, 2007; Ando and Oka, 2021; Pöppelmeier et al., 2021). Most of the hysteresis experiments have been performed with FWF in the latitudinal belt between 20–50°N in the Atlantic, thereby avoiding a direct perturbation of the convection sites further north in order to focus on the salt-advection feedback. FWF applied in the convection areas has a stronger impact on the AMOC (e.g. Ganopolski and Rahmstorf, 2001; Smith and Gregory, 2009), because the state of the AMOC is tightly linked to the production of deep water.

Convection and deep water formation do not only depend on surface freshwater flux, but are more generally controlled by the surface buoyancy flux, which also depends on the net heat losses and the temperature at the sea surface. The temperature dependence of the surface buoyancy flux arises from the nonlinear equation of state of seawater, and in particular from the temperature dependence of the thermal expansion coefficient (e.g. Roquet et al., 2015). Perturbations to the climate will affect both the net surface freshwater flux, as a result of changes in the hydrological cycle, and the surface temperature, with intricate implications for AMOC stability. The effect of climate on AMOC stability has been investigated in relatively few studies, mainly by changing the concentration of atmospheric $CO_2$ (e.g. Brown and Galbraith, 2016; Klockmann et al., 2018; Galbraith and de Lavergne, 2019). These studies show a generally stronger AMOC in equilibrium with higher $CO_2$, but mostly focused on climates colder than present. Recently, Gérard and Crucifix (2024) have performed model simulations with slowly increasing and decreasing $CO_2$, producing an AMOC hysteresis in $CO_2$ space and suggesting an AMOC weakening in equilibrium with a warmer climate.

For an improved understanding of past and future AMOC evolution it is important to consider changes in climate and changes in the surface ocean freshwater balance due to changing land ice volume, since both play an important role for AMOC stability. Here we use an Earth system model to systematically explore the combined effect of surface FWF and climate on AMOC stability. The effect of external FWF is quantified by running experiments with FWF in different latitudinal belts in the North Atlantic, while the effect of climate is explored by varying the atmospheric $CO_2$ concentration, which is one of the main factors driving past and future climate changes.

## 2 AMOC hysteresis in freshwater space

A common approach to investigate the stability of the AMOC is to apply a slowly changing perturbation in the surface fresh-water balance of the North Atlantic (Stocker and Wright, 1991; Rahmstorf, 1995). We used the fast Earth system model CLIMBER-X (Willeit et al., 2022) to perform standard FWF experiments to track the stable states of the AMOC. CLIMBER-X has a horizontal resolution of 5°x5°in the atmosphere, ocean, sea ice and land components and 23 unequally-spaced vertical layers in the ocean (see Appendix A1) and has been shown to perform well both in terms of present-day simulated climate and in terms of sensitivities to different forcings and changes in boundary conditions (Willeit et al., 2022). Notably, the model has recently been shown to reproduce Dansgaard-Oeschger events under mid-glacial conditions (Willeit et al., 2024), further confirming that it is a suitable tool to study AMOC stability. CLIMBER-X is a computationally efficient model that allows to perform the long simulations required for a comprehensive stability analysis of the AMOC.

The FWF, as used in this study, represents perturbations to the freshwater balance of the North Atlantic by factors external to the climate (atmosphere-ocean-sea ice-land) system, namely from changing land ice volume that is not accounted for in our simulations because we use prescribed present-day ice sheets. When driven by slowly varying changes in the FWF in different latitudinal belts in the North Atlantic (see Appendix A2), CLIMBER-X shows the typical hysteresis behaviour (Fig. 1) seen also in a hierarchy of other models of varying complexity (Ganopolski and Rahmstorf, 2001; Rahmstorf et al., 2005; Hofmann and Rahmstorf, 2009; Hawkins et al., 2011; Hu et al., 2012; Ando and Oka, 2021; van Westen and Dijkstra, 2023; Gérard and Crucifix, 2024). In particular there is a range of FWF, roughly between 0.01 and 0.17-0.18 Sv, over which the AMOC has two stable states. The AMOC in the model is monostable under pre-industrial conditions (*Modern* AMOC state), although relatively close to bi-stability, as the *Off* AMOC state is also stable for FWF>0.01 Sv. The hysteresis is wider if FWF is applied between 20–50°N compared to when it is applied to the latitudinal belt 50–70°N, where convection occurs (Fig. 1). When FWF is increased at 280 ppm of $CO_2$, a critical point is reached where the AMOC shows a rather abrupt (both in FWF space and in time) weakening, indicating a prominent role of convective instability as opposed to the expected parabolic shape resulting from a Stommel-like bifurcation (Stommel, 1961). The AMOC is more sensitive to FWF perturbations applied between 50°N and 70°N, in which case an abrupt weakening of the AMOC occurs already for a hosing of ∼0.05 Sv (Fig. 1) leading to a transition into a *Weak* AMOC state. This is the result of a collapse of deepwater formation in the Labrador and Irminger Seas (Fig. B1) and a general shift of the convection to latitudes south of ∼55°N, resembling Stadial-like conditions of past Dansgaard-Oeschger (DO) events (Willeit et al., 2024). The associated 'overshoot' in Fig. 1 is the result of a damped oscillation caused by the crossing of the bifurcation point between *Modern* and *Weak* AMOC states. In the presence of noise, such oscillations can become quasi-periodic, as shown in Willeit et al. (2024).

It should be noted that the *Weak* AMOC state is seen only in the experiments with FWF hosing applied directly to the convection regions between 50°N and 70°N. Most AMOC hysteresis experiments to date have been performed with FWF at lower latitudes (usually between 20°N and 50°N e.g. Rahmstorf et al., 2005; Hu et al., 2012; van Westen and Dijkstra, 2023)). In the model, a complete AMOC shutdown occurs for FWF of ∼0.17-0.18 Sv, depending on the latitude of the applied forcing (Fig. 1). When FWF is then slowly decreased again, the AMOC recovers from the *Off* state with an overshoot at

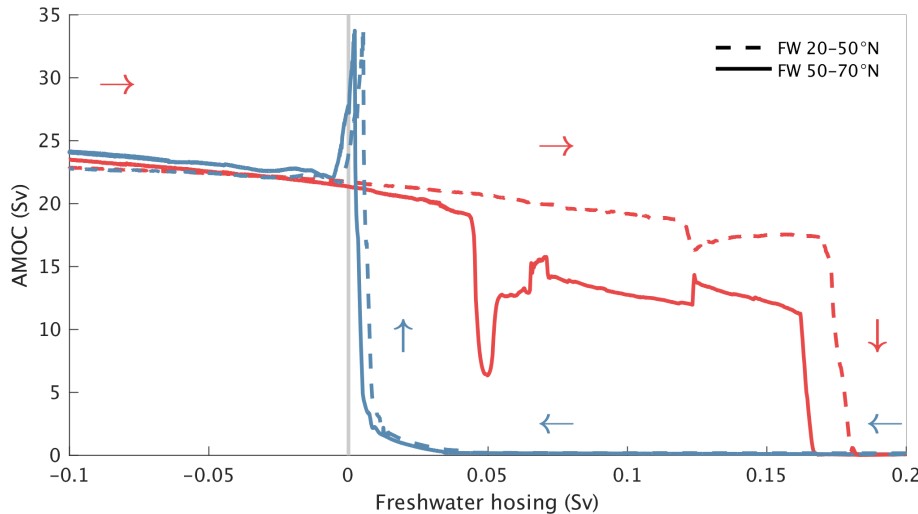

**Figure 1.** Hysteresis of the AMOC in freshwater space. AMOC response to prescribed changes in FWF in two different latitudinal belts in the North Atlantic (dashed lines for 20–50°N and solid lines for 50-70°N). The red lines are from simulations with increasing FWF starting at -0.5 Sv and the blue lines are for experiments with decreasing FWF starting at +0.5 Sv. In all cases the rate of change of the imposed FWF is $0.02\,\text{Sv}\,\text{kyr}^{-1}$, so that each full simulation covering the FWF range between -0.5 Sv to +0.5 Sv (only partly shown in the figure) corresponds to 50,000 simulation years. The AMOC strength is defined as the maximum of the Atlantic meridional streamfunction deeper than 700 m.

$\approx$0.01 Sv, independently from the latitude at which the FWF is applied. The overshoots are a result of the transient nature of our experiments and become less prominent with slower rates of FWF changes (Fig B2).

In our freshwater hysteresis experiments in Fig. 1 we applied a very slow rate of change of 0.02 Sv per 1000 years in the FWF. When repeating the experiment with a ten times higher rate of change ($0.2\,\text{Sv}\,\text{kyr}^{-1}$), a value typically used in
computationally expensive state-of-the-art climate models (e.g. Hu et al., 2012; van Westen and Dijkstra, 2023), the hysteresis looks very different (Fig. B2). For a higher rate of change in the forcing, the hysteresis is generally smoother and more regular and does not show the abrupt transitions that characterize the hysteresis curves produced with slow FWF changes. Notably, the *Weak* AMOC mode is not captured in the fast hysteresis experiments, where the AMOC is gradually transitioning to an *Off* state when the FWF exceeds $\sim$0.05 Sv (Fig. B2b). In the experiments with decreasing FWF a higher forcing rate leads to
a much delayed recovery of the AMOC from the *Off* state, resulting in a very distorted representation of the bistability range. In particular, the slow forcing experiments show that the AMOC is monostable under pre-industrial conditions in our model, while the fast forcing simulations give the wrong impression that the AMOC *Off* state is also stable (Fig. B2).

## 3 Equilibrium AMOC response to $CO_2$ changes

Another way of looking at AMOC stability is to investigate the AMOC response to changes in atmospheric $CO_2$. In Fig. 2
the $CO_2$ concentration is very slowly increased starting from 180 ppm up to 560 ppm ($0.002\,\%\,yr^{-1}$, see also Appendix A2),
the model shows a general increase in AMOC strength with increasing global temperature under quasi-equilibrium conditions
(Fig. 2, red line). A weaker AMOC for $CO_2$ concentrations lower than pre-industrial has been found also in general circulation
models (Stouffer and Manabe, 2003; Oka et al., 2012, 2021; Brown and Galbraith, 2016; Klockmann et al., 2018; Galbraith and
de Lavergne, 2019). Stouffer and Manabe (2003) found AMOC strengthening under doubling and quadrupling of $CO_2$ relative
to pre-industrial levels, and recently Bonan et al. (2022) showed that at least some state-of-the-art climate models produce an
AMOC that is appreciably stronger under $CO_2$ quadrupling. Gérard and Crucifix (2024) recently analyzed the AMOC response
to a slow $CO_2$ increase and found a gradual AMOC weakening and eventual collapse at $CO_2$ above $\sim$1500 ppm. This is in
contrast to our results, which show an increase in AMOC strength with increasing $CO_2$, at least up to a $CO_2$ concentration of
560 ppm. It should be noted that both in Gérard and Crucifix (2024) and in our study the $CO_2$ increase is slow enough to track
the equilibrium AMOC response.

The general AMOC strenghtening with warming in the model is punctuated by two abrupt transitions at $\sim$250 ppm and
$\sim$370 ppm (Fig. 2), separating three different AMOC states and convection patterns in the North Atlantic (Fig. 3). The different
AMOC states are formally defined based on a critical depth of the maximum mixed layer depth in three different regions in
the northern North Atlantic as shown in Fig. B3. A stronger AMOC is generally associated with a northward shift of the sites
of deepwater formation (Fig. 3f-h), following the northward retreat of sea ice (Fig. 3b-d). A few general circulation models
found thermal AMOC thresholds under climate conditions generally colder than the pre-industrial, leading to abrupt AMOC
weakening when climate is cooled and abrupt AMOC strengthening when climate is warmed (Knorr and Lohmann, 2007;
Banderas et al., 2012; Oka et al., 2012; Zhang et al., 2017). Adloff et al. (2024) also found several thermal thresholds in the
AMOC in idealized simulations of the last glacial cycle.

Willeit et al. (2024) have shown that around the transition between a *Modern* and *Weak* AMOC at $\sim$250 ppm CLIMBER-X
simulates Dansgaard-Oeschger-like events in the presence of noise, even with modern ice sheets. This millennial-scale variability originates from internal climate system dynamics associated with transitions between two distinct convection patterns that
are stable for different $CO_2$ concentrations. For $CO_2$ above $\sim$250 ppm, the convection pattern resembles the present-day state
with deep water forming in the Labrador Sea and in the Nordic Seas (Fig. 3c), while for $CO_2$ below $\sim$250 ppm convection
can not be sustained in the Labrador and Irminger Seas and is generally restricted to areas south of $\sim$55°N (Fig. 3b). This state
is equivalent to the *Weak* AMOC state in Fig. 1. A narrow window of $CO_2$ concentrations exists for which both convection
patterns are stable for the same $CO_2$ (Fig. 2, red vs blue solid lines), but in Willeit et al. (2024) it was shown that this bistability
is not a requirement for the existence of millennial-scale variability, for which it is sufficient that the system is close enough to
the bifurcation point between *Modern* and *Weak* AMOC states. It should be noted that in the past $CO_2$ concentrations below
the pre-industrial level of 280 ppm were related to the appearance of continental ice sheets over the NH, which affect AMOC
stability (e.g. Zhang et al., 2014; Klockmann et al., 2018; Willeit et al., 2024) but are not considered in the present study.

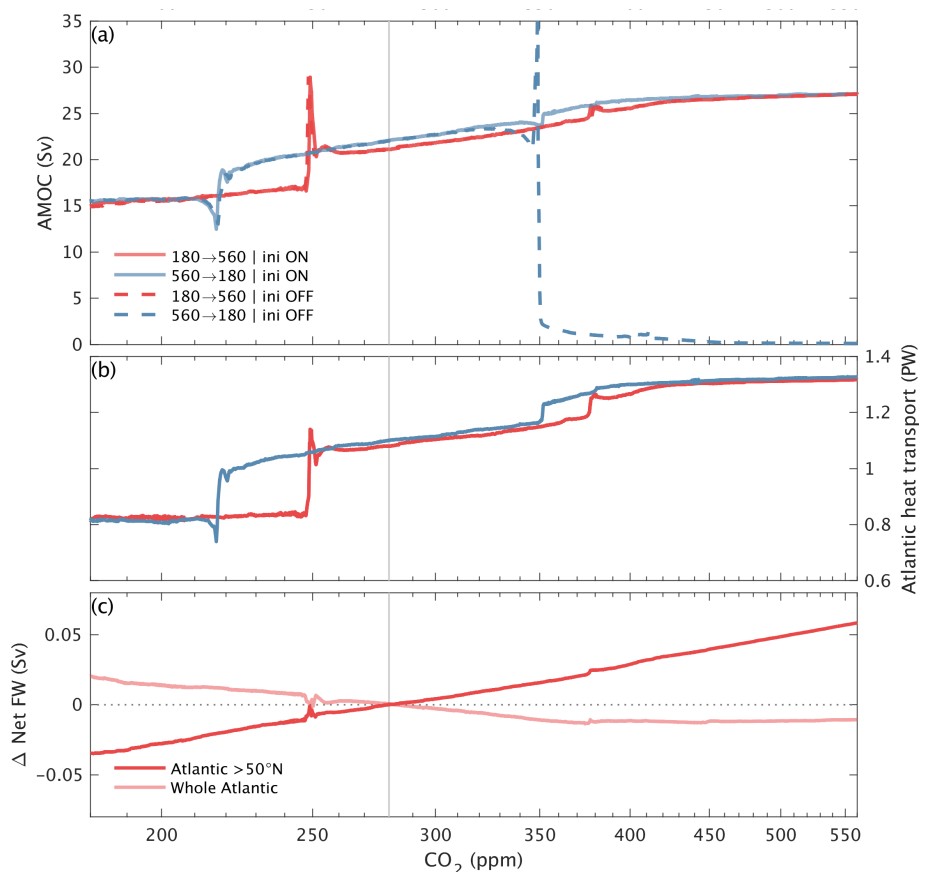

**Figure 2.** Quasi-equilibrium AMOC response to changes in $CO_2$. (a) Maximum of the AMOC streamfunction deeper than 700 m, (b) maximum meridional heat transport by the ocean in the Atlantic and (c) changes in net surface freshwater flux in the Atlantic (dark red line for the area >50°N and light red line for the whole Atlantic ocean) in simulations with slowly varying prescribed atmospheric $CO_2$ concentrations for $CO_2$ increasing from 180 ppm (red lines) and for $CO_2$ decreasing from 560 ppm (blue lines). The solid lines are from simulations initialized form a pre-industrial AMOC state, while the dashed lines are from simulations initialized from the AMOC off state. In (b) and (c) only selected simulations are shown. The solid vertical line indicates the pre-industrial $CO_2$ concentration of 280 ppm.

The AMOC transition at ∼370 ppm is associated with a convection start in the Kara Sea and Nansen Basin (Fig. 3c), and has a clearer imprint in the Atlantic meridional heat transport (Fig. 2b) than in the maximum strength of the the AMOC (Fig. 2a). We term this the *Strong* AMOC state. Convection in the Arctic is triggered in several climate models in response to future transient global warming (Brodeau and Koenigk, 2016; Lique et al., 2018; Lique and Thomas, 2018; Bretones et al., 2022; Pan et al., 2023), although in many other models this does not occur (Heuzé and Liu, 2024). However, the fact that convection in the Arctic in some models starts even in transient simulations that are characterized by an overall weakening of the AMOC, suggests that the Arctic would be a plausible new location of deep water formation in a warmer climate under quasi-equilibrium conditions.

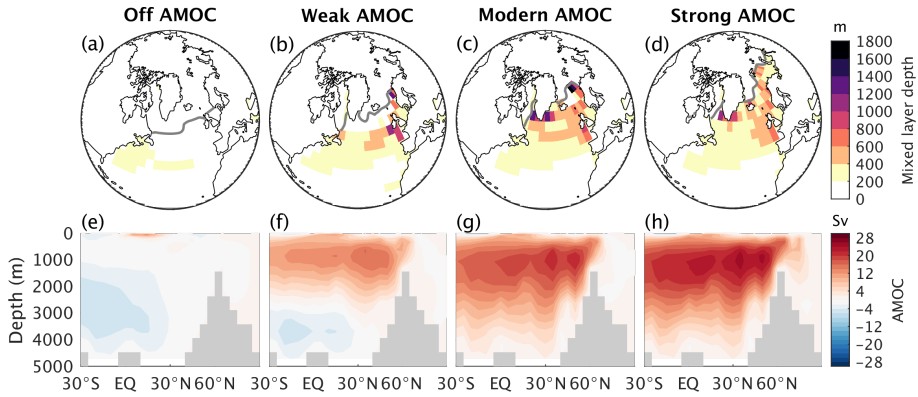

**Figure 3.** The different AMOC states. (a-d) Maximum monthly mean mixed layer depth of the year and (e-h) AMOC streamfunction for the different equilibrium AMOC states in the model: (a,e) *Off*, (b,f) *Weak*, (c,g) *Modern* and (d,h) *Strong* AMOC states. The grey line in (a-d) shows the maximum monthly mean sea ice extent of the year (defined as sea ice concentration >0.15). The shown AMOC states are all for the same boundary conditions of 400 ppm of atmospheric $CO_2$ and a FWF of +0.05 Sv, i.e. for the boundary conditions under which all four AMOC states co-exist in the model (Fig. 5).

Model simulations initialized at 180 ppm with an *Off* AMOC state (see Appendix A2) (Fig. 2a, dashed red line) indicate that the AMOC *Off* state is not stable for $CO_2$ concentrations below pre-industrial levels, resulting in a monostable AMOC for pre-industrial conditions in our model (excluding the relatively narrow hysteresis around the transition between 220 and 250 ppm). However if the model is initialized in an AMOC *Off* state at 560 ppm the AMOC remains in the *Off* state as long as $CO_2$ decreases to values below $\sim$350 ppm, after which the AMOC recovers with an overshoot (Fig. 2a, dashed blue line).

Our model has therefore two widely different stable AMOC states for $CO_2$ concentrations above $\sim$350 ppm, the AMOC *Off* state and a *Strong* state characterized by a more vigorous AMOC than *Modern* states (Fig. 2a). Hu et al. (2023) performed a similar AMOC hysteresis analysis using future climate scenarios, and found that the AMOC exhibits two stable states for $CO_2$ concentrations $\sim$1000 ppm in their model.

It is interesting to note that the AMOC strengthening with global warming occurs despite an associated increase in the

net surface freshwater flux into the northern North Atlantic, north of 50°N (Fig. 2c). This is a result of an intensification of the hydrological cycle in a warming climate, with the typical wet-gets-wetter and dry-gets-drier pattern (Held and Soden, 2006; Zhang et al., 2013). For double the amount of $CO_2$ the net freshwater flux into the northern North Atlantic increases by $\sim$0.07 Sv, a relatively large freshwater flux, which is approximately an order of magnitude higher than the net freshwater flux from the Greenland ice sheet simulated under similar temperatures (e.g. Calov et al., 2018; Briner et al., 2020), and would

roughly correspond to the rate of freshwater input resulting from the Greenland ice sheet melting completely over a time period of $\sim$1500 years. The increase in freshwater at high latitudes in the Atlantic as a response to global warming is a consistent feature also of CMIP6 models under transient future scenarios (Fig. B4a). While the northern North Atlantic gets wetter as climate warms, the net surface freshwater flux into the whole Atlantic Ocean shows the opposite trend in our model, with a small decrease in net freshwater flux as $CO_2$ concentrations increase (Fig. 2c). Most CMIP6 models show a larger decrease

of the net freshwater flux into the Atlantic than CLIMBER-X as climate warms (Fig. B4b), but with a relatively wide spread between models.

The effect of changes in the net surface freshwater flux associated with global warming on AMOC stability is therefore the result of two competing effects: (i) salinification of the Atlantic as a whole, which stabilizes the AMOC through the salt-advection feedback, and (ii) the freshening of the northern Atlantic region, which destabilizes the AMOC through an increased 170 surface buoyancy flux and a consequent lowering of the surface seawater density in the deep-water formation regions.

## 4 AMOC stability landscape in combined $CO_2$ and freshwater space

The above analyses of the response of the AMOC to changes in FWF and atmospheric $CO_2$ have shown that there are at least four different stable AMOC states in CLIMBER-X, namely *Off*, *Weak*, *Modern* and *Strong*. The $CO_2$–freshwater conditions under which these different AMOC states are stable can be investigated by tracing the AMOC response in the $CO_2$–FWF 175 space. This is done by slowly following the $CO_2$–FWF paths illustrated in Fig. A2 as described in detail in Appendix A2. The standard approach to tracing the AMOC stability diagram is to slowly change one of the control parameters (usually FWF) first in one direction and then in the opposite direction. However, such a method often fails to trace all equilibria, especially when more than two equilibrium states coexist at the same point in the phase space. Therefore, we combined the traditional approach, which works in our case for the *Strong* and *Off* modes, with a more sophisticated procedure where we alternate 180 changes in FWF and $CO_2$ space to trace the *Modern* and *Weak* states (Fig. A2). The results are shown separately for each AMOC state in Fig. 4. High $CO_2$ and low FWF generally favor stronger AMOC states (Fig. 4 and Fig. 5). The *Strong* AMOC state is characteristic of climates warmer than pre-industrial (Fig. 4d). As seen already in Fig. 2, without FWF the AMOC transitions to the *Strong* state for $CO_2$ concentrations above ∼380 ppm. The *Modern* AMOC state covers conditions going from low $CO_2$ and negative FWF to high $CO_2$ and FWF up to 0.1 Sv, passing through pre-industrial conditions (Fig. 4c). If 185 the climate would be in equilibrium with present-day $CO_2$ concentrations of ∼420 ppm, the model suggests that the *Modern* AMOC state would not be stable, but that the AMOC would rather be in the *Strong* state instead (Fig. 4c,d). The *Weak* AMOC state exists for a range of $CO_2$ concentrations between ∼200 and ∼560 ppm and FWF roughly between -0.05 and 0.18 Sv (Fig. 4b). Starting from pre-industrial conditions the *Weak* AMOC state can be reached either by reducing $CO_2$ or by adding freshwater to the North Atlantic, north of 50°N, as shown also in Fig. 2a,b and Fig. 1. For the investigated range of $CO_2$ 190 concentrations an *Off* AMOC state can not be achieved by varying $CO_2$ alone, but only through a large enough FWF. Under quasi-equilibrium conditions, the FWF needed to shut down the AMOC when starting from an 'on' AMOC state is in the range ∼0.05–0.2 Sv, depending on the $CO_2$ concentration (Fig. 4b and Fig. 5). If the FWF is larger than ∼0.05 Sv, the *Off* AMOC state is stable for any $CO_2$ concentration (Fig. 4a). For smaller or negative FWF the stability of the *Off* state depends on $CO_2$. The AMOC bistability range generally broadens with warming (Fig. 5), particularly because the AMOC recovery from the 195 *Off* state requires an increasingly more negative FWF (Fig. 4a). In the model the *Off* state is not stable under pre-industrial conditions, but it is for higher $CO_2$ concentrations (Fig. 4a).

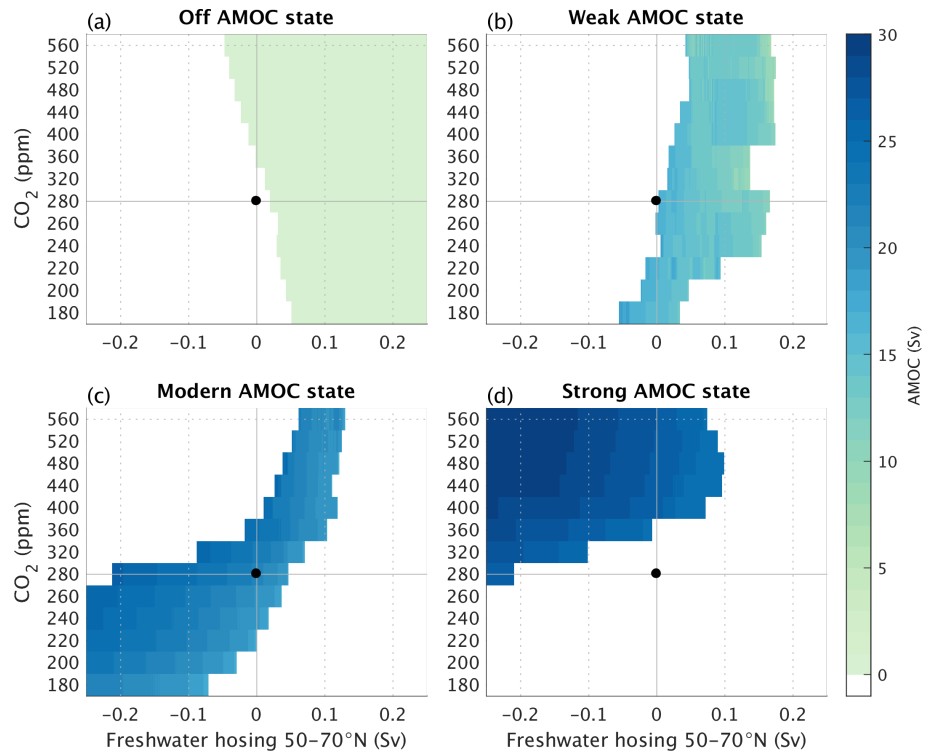

**Figure 4.** AMOC states in combined $CO_2$ and freshwater space. Maximum of the AMOC streamfunction as a function of $CO_2$ and FWF between 50–70°N separately for the four different stable AMOC states in the model, namely (a) *Off* AMOC state, (b) *Weak* AMOC state, (c) *Modern* AMOC state and (d) *Strong* AMOC state. The different states are formally defined based on a critical threshold ($mld_{max}^{crit}$ = 600 m) of the maximum mixed layer depth ($mld_{max}$) in three separate regions in the North Atlantic, namely (i) the Nordic Seas, (ii) the Labrador Sea and (iii) the Barents and Kara Seas and the Nansen basin. *Off*: $mld_{max} < mld_{max}^{crit}$ in (i-iii); *Weak*: $mld_{max} > mld_{max}^{crit}$ in (i) and $mld_{max} < mld_{max}^{crit}$ in (ii-iii); *Modern*: $mld_{max} > mld_{max}^{crit}$ in (i-ii) and $mld_{max} < mld_{max}^{crit}$ in (iii); *Strong*: $mld_{max} > mld_{max}^{crit}$ in (i-iii). The black dot indicates pre-industrial conditions. The stability landscape is constructed based on simulations following the paths shown in Fig. A2, where the rate of change of $CO_2$ is 0.002 % yr$^{-1}$ as in Fig. 2 and the rate of change of FWF is 0.02 Sv kyr$^{-1}$ as in Fig. 1.

We use the remarkable fact that all four AMOC states in the model are stable under the same boundary conditions for $CO_2$ concentrations ∼440 ppm and FWF ∼0.05 Sv (Fig. 5) to isolate the effect of the changes of AMOC states on the climate. An AMOC weakening generally causes a cooling that is most pronounced in the northern North Atlantic, but that also extends more widely to the mid- to high-latitudes of the Northern Hemisphere (Fig. 6), with the largest effect being observed in winter (Fig. 6e-h) and the weakest in summer (Fig. 6i-l). This is in general agreement with previous studies looking at the climate impact of an AMOC shutdown forced by FWF (e.g. Jackson et al., 2015; van Westen and Dijkstra, 2023). The largest possible effect of AMOC on climate is for a transition between the *Strong* and *Off* states (Fig. 6a,e,i), which shows an annual mean cooling of up to 20°C in the northern North Atlantic, with temperatures as much as ∼25-30°C colder in winter, associated also with a pronounced sea ice advance (Fig. 6e). A shift from *Strong* to *Modern* AMOC induces a cooling of ∼10°C in

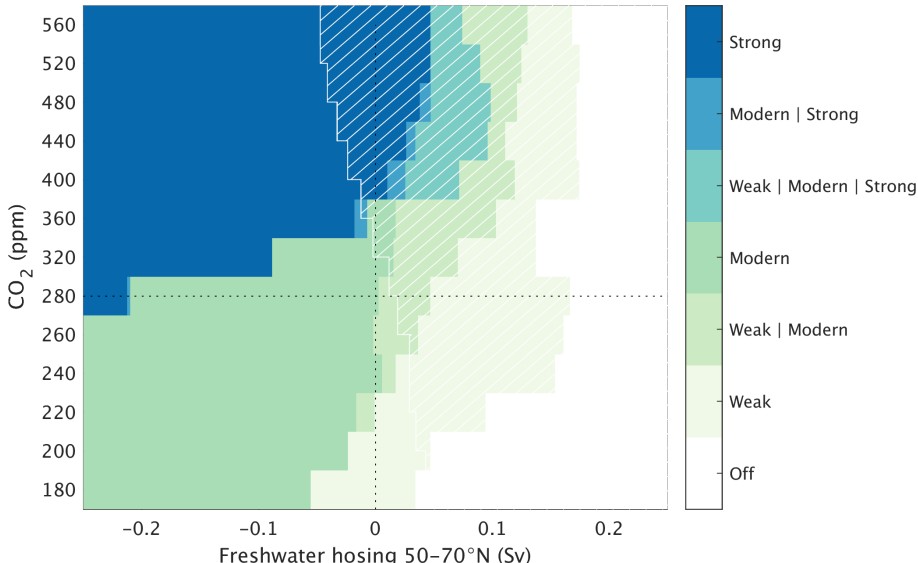

**Figure 5.** Summary of AMOC stability landscape in combined $CO_2$ and freshwater space. The colored regions indicate the on AMOC states that are stable under the given $CO_2$ and FWF as indicated in the legend. The filled white area indicates where only the *Off* AMOC state is stable, while the white hatched area shows the domain where the *Off* AMOC state and one or more of the three 'on' AMOC states coexist. Note that multiple AMOC states are stable under some boundary conditions.

the Barents and Kara Seas (Fig. 6b,f,j). A slow increase in the $CO_2$ concentration, which would trigger a *Modern* to *Strong* AMOC transition as shown in Fig. 2a, would therefore cause a warming in these regions, with the opposite sign of changes in Fig. 6b,f,j. A transition from *Modern* to *Weak* AMOC mainly affects the Nordic Seas with a cooling by up to $\sim$15°C in winter (Fig. 6c,g,k). A shift from a *Weak* to an *Off* AMOC state has a strong imprint on temperatures in the Nordic Seas and the Labrador and Irminger Seas (Fig. 6d,h,l). In general, the differences in climate between the different AMOC states are related to changes in ocean heat transport and the shifts in the location of deep water formation in the North Atlantic and the associated changes in sea ice extent (Fig. 3a-d). While the temperature differences in Fig. 6 are representative of the impact of the different AMOC states on climate, they are strictly valid only for an atmospheric $CO_2$ of $400\,\mathrm{ppm}$ and a freshwater forcing of $0.05\,\mathrm{Sv}$ and could differ if the transition between AMOC states occurs under different boundary conditions.

## 5 Discussion and Conclusions

For the first time, we have performed a systematic analysis of the AMOC stability in the FWF–$CO_2$ space. This was done by very slowly varying the surface freshwater flux in the North Atlantic and the atmospheric $CO_2$ concentration and required $\sim$1,000,000 model years of simulation.

We found four distinct modes of the AMOC. Apart from an *Off* AMOC state with no North Atlantic deep water formation and a *Modern*-like AMOC state with deep water forming in the Labrador and Nordic Seas as observed at present, we find two

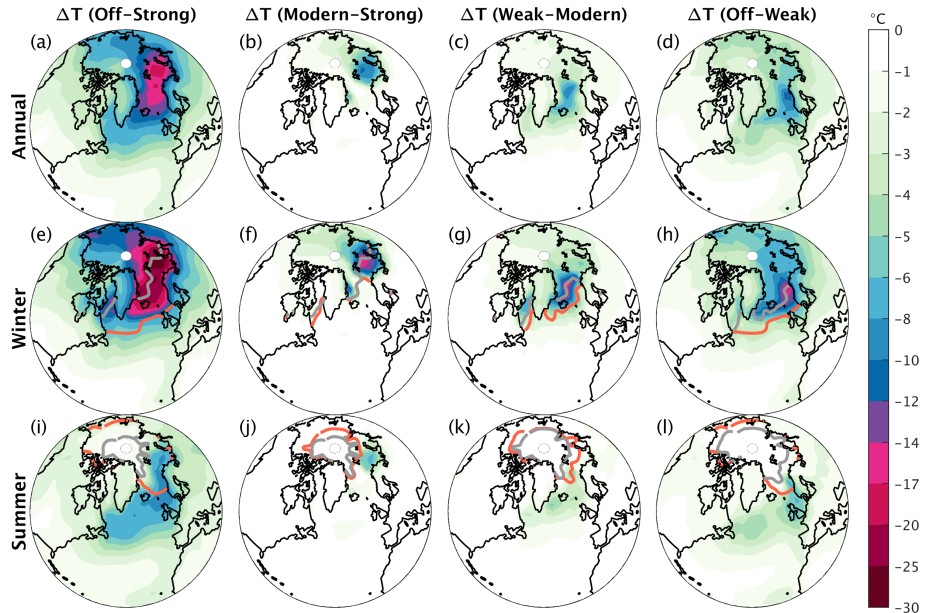

**Figure 6.** Temperature differences induced by different AMOC states. (a-d) Annual, (e-h) winter (December-January-February) and (i-l) summer (June-July-August) temperature differences between different AMOC states as indicated in the panels. The colored lines show the maximum monthly mean sea ice extent of the year in (e-h) and the seasonal minimum sea ice extent in (i-l), with the grey line always corresponding to the stronger AMOC state and the orange line to the weaker AMOC state. Sea ice extent is defined as sea ice concentration >0.15. The figure shows temperature differences between the different AMOC states for 400 ppm of atmospheric $CO_2$ and a FWF of +0.05 Sv, i.e. for the boundary conditions under which all four AMOC states co-exist in the model (Fig. 5).

additional equilibrium states: (i) a *Weak*, Stadial-like, AMOC state with deep water forming predominantly south of ∼55°N, (ii) a *Strong* AMOC state with convection reaching into the Arctic for $CO_2$. Intermediate stable AMOC states between *Modern* and *Off* associated with changes in the convection pattern have also been found in previous studies (Rahmstorf, 1995; Lohmann et al., 2024) using ocean-only models forced with increasing FWF, but our results indicate that the standard method of tracing hysteresis in the FWF space may not be enough to find all possible AMOC modes.

Our AMOC stability landscape demonstrates that interglacial climates of the Quaternary are generally stable, because of the mono-stability of the AMOC under pre-industrial-like conditions. The fact that the AMOC is monostable for $CO_2$ concentrations around 280 ppm, a typical value for interglacials, in the absence of FWF also explains why the AMOC always recovered at the end of glacial terminations, after temporary shutdowns induced by the freshwater input (∼0.1 Sv) from rapidly melting ice sheets. The existence of the *Weak* AMOC state has been shown by Willeit et al. (2024) to be related to Dansgaard-Oeschger events in the model, explaining the large AMOC variability observed during glacial times. Our results suggest that a different mode of AMOC (the *Strong* state) was possible during past warm climate conditions. The Pliocene was the most recent period in Earth's history with elevated atmospheric $CO_2$ concentrations of ∼400 ppm (Martínez-Botí et al., 2015; Seki et al., 2010), which, according to our results, would be high enough to push the AMOC to a *Strong* state. There is indeed proxy-based

evidence of a stronger-than-present AMOC in the Pliocene (Raymo et al., 1996; Ravelo and Andreasen, 2000) with an increased northward ocean heat transport in the Atlantic (Dowsett et al., 1992), which is consistent with sea surface temperature reconstructions for this period showing warmer conditions in the North Atlantic (McClymont et al., 2020). Climate models also tend to produce a stronger AMOC under mid-Pliocene conditions, although with considerable spread (Zhang et al., 2021; Weiffenbach et al., 2023). Whether the existence of the *Strong* AMOC state could potentially lead to some kind of centennial

or millenial-scale variability in the AMOC in a warmer climate remains to be explored.

As long as freshwater input from melting ice sheets is small, our results indicate a generally stronger and deeper AMOC at equilibrium under warmer climate conditions. This does not contradict to the projected AMOC weakening response to anthropogenic global warming (e.g. Bellomo et al., 2021; Weijer et al., 2020; Weaver et al., 2012), which is an intrinsically transient response of the system predominantly induced by the rapid temperature increase (Gregory et al., 2005; Weaver et al.,

2007; Levang and Schmitt, 2020). For present-day conditions and even up to the highest considered $CO_2$ concentration of 560 ppm, the net freshwater flux from Greenland is small ($\ll 0.1$ Sv (Otosaka et al., 2023; Calov et al., 2018; Briner et al., 2020) and has therefore little effect on the AMOC, as also indicated by coupled climate–ice sheet model simulations (Bakker et al., 2016; Ackermann et al., 2020).

In the phase space, a $CO_2$ increase drives the AMOC towards a stronger state. This is because the AMOC response to $CO_2$

is fundamentally different from the response to FWF. In the case of FWF, regardless of the rate of change, an increase in FWF weakens the AMOC. In the case of $CO_2$, this is not true: a fast enough increase in $CO_2$ weakens the AMOC (Stocker and Schmittner, 1997), while a very slow (quasi-equilibrium) increase strengthens it. Since the main cause of future AMOC weakening is the increase in $CO_2$, the traditional FWF hysteresis analysis is of limited use for predicting the future AMOC evolution. The non-trivial relation between future projected AMOC evolution and the stability landscape will be the subject of

future work.

Even if our stability diagram can not explain the AMOC response to transient $CO_2$ forcing, it provides some information on whether the transient weakening of the AMOC is reversible (mono-stable regime) or irreversible (bi-stable). Our results suggest that a future AMOC shutdown, which could be triggered by the transient response to anthropogenic global warming, could be irreversible because the *Off* AMOC state is stable for $CO_2$ concentrations above the present-day level. Our model

simulations therefore indicate that in terms of stability landscape the AMOC is currently moving towards a stronger state, but from a monostable into a bistable regime, where the AMOC *Off* state is also stable. It is hence in principle possible that slightly different future global warming trajectories could lead in one case to an irreversible (on multi-centennial time scales) AMOC shutdown and in another case to a transient AMOC weakening followed by a transition into a *Strong* AMOC state, eventually resulting in fundamentally different climate conditions in the North Atlantic. Transient model simulations under

future emission scenarios will have to be performed to explore this possibility.

It should be noted that the AMOC stability landscape presented above is a result of model simulations with the fast Earth system model CLIMBER-X, which has a relatively coarse-resolution and whose ocean model is based on the quasi-geostrophic approximation, with all the attendant limitations. Generally, anything related to convection and changes in convective patterns is highly model dependent, with widely different results produced even among state-of-the-art general circulation models

(Sgubin et al., 2017; Heuzé, 2017; Treguier et al., 2023). Obviously, this does not question the existence of distinct Stadial and Interstadial AMOC modes during glacial times. Since the experiments presented in the paper can only be performed with a model like CLIMBER-X, we believe that they are useful to illustrate the general concept of AMOC stability. CLIMBER-X does not produce internal interannual climate variability and it is possible that different modes of the AMOC, which are distinct in our simulations, may not be distinguished in the presence of strong variability (e.g. Monahan, 2002). The presence of noise

can also lead to spontaneous transitions between different AMOC modes, as demonstrated in Willeit et al. (2024). Intrinsic internal variability in general circulation models can make the tracing of the AMOC stability landscape problematic, and in this sense, the absence of such variability in CLIMBER-X is actually an advantage of the model, since it allows to obtain the phase portrait of the system without noise and then to simulate realistic dynamical behaviour of the system by adding the noise.

*Code and data availability.* The CLIMBER-X model is freely available as open source code at https://github.com/cxesmc/climber-x.

**Appendix A: Materials and Methods**

**A1 Earth System Model**

We use the CLIMBER-X Earth system model (Willeit et al., 2022) in a climate-only setup, including the frictional-geostrophic 3D ocean model GOLDSTEIN (Edwards et al., 1998; Edwards and Marsh, 2005) with 23 vertical layers, the semi-empirical statistical-dynamical atmosphere model SESAM (Willeit et al., 2022), the dynamic-thermodynamic sea ice model SISIM

(Willeit et al., 2022) and the land surface model with interactive vegetation PALADYN (Willeit and Ganopolski, 2016). All components of the climate model have a horizontal resolution of $5°x5°$. Ice sheets are prescribed at their modern state and the net FWF from ice sheets is zero. The model is open source, is described in detail in Willeit et al. (2022) and in general shows performances that are comparable with state-of-the-art CMIP6 models under different forcings and boundary conditions. In particular, the simulated present-day AMOC overturning profile at $26°N$ in the Atlantic is close to observations (Fig. A1),

although it reaches a bit too deep. The present-day deep convection patterns compare well to ocean reanalysis in the North Atlantic (Fig. 13 in Willeit et al., 2022).

**A2 Experiments**

With CLIMBER-X we ran transient simulations where we slowly varied either the FWF in the North Atlantic or the atmospheric $CO_2$ concentration.

The standard FWF experiments were performed with prescribed changes in freshwater flux with a rate of $\pm0.02\,\mathrm{Sv\,kyr^{-1}}$ in two different latitudinal belts, 20–50°N and 50–70°N in the North Atlantic, either starting from an initial hosing flux of -0.5 Sv and slowly increasing it until 0.5 Sv, or starting from +0.5 Sv hosing flux and gradually decreasing it until -0.5 Sv. Each simulation is 50,000 years long. These two experiments were performed with prescribed constant $CO_2$ concentration of

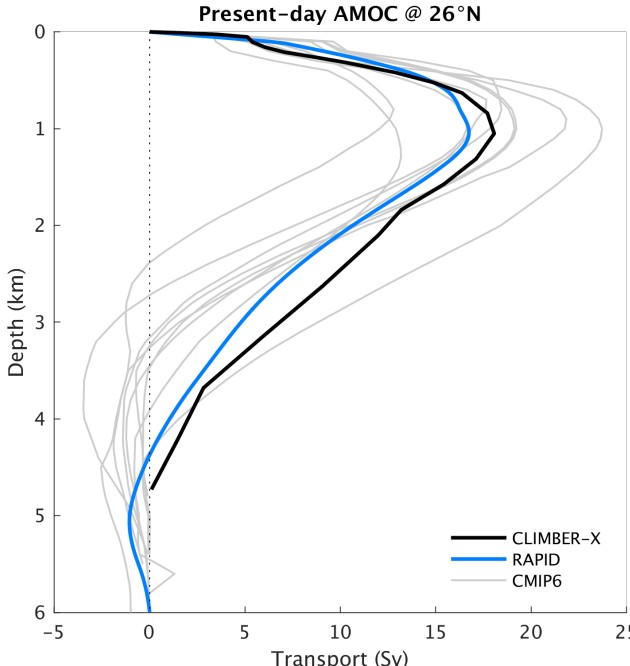

**Figure A1.** Vertical profile of the simulated Atlantic meridional overturning streamfunction at $26°N$ (black) compared to observations from the RAPID array (Frajka-Williams et al., 2019) (blue) and a selection of CMIP6 models (grey). The CLIMBER-X and CMIP6 streamfunction is computed from historical simulations as the average over the time period from 2000 to 2014, while the RAPID values represent an average from 2004 to 2020.

280 ppm. The initial condition for both these experiments is a pre-industrial equilibrium simulation run for 10000 years with
280 ppm of $CO_2$ and present-day ice sheets. To investigate how the rate of change of the FWF affects the hysteresis behaviour, we also repeated the freshwater hysteresis analysis using a $10\times$ faster rate of change of the FWF ($0.2\,\mathrm{Sv\,kyr^{-1}}$) and a slower rate of change of $0.005\,\mathrm{Sv\,kyr^{-1}}$).

We additionally performed transient simulations with slowly varying $CO_2$ concentrations: (i) starting at 180 ppm and gradually increasing $CO_2$ up to 560 ppm and (ii) starting from 560 ppm and gradually decreasing $CO_2$ down to 180 ppm. In both
cases the rate of change of $CO_2$ is $2\,\%\,\mathrm{kyr^{-1}}$ implying a total simulation length of $\sim$56,500 years. We have chosen an exponential $CO_2$ change rate in order to get a roughly linear global temperature response with time, considering the logarithmic dependence of the $CO_2$ radiative forcing. The initial state for these simulations is a 10,000 years long equilibrium run with either 180 ppm (for (i)) or 560 ppm (for (ii)) of atmospheric $CO_2$. Simulations (i) and (ii) are also repeated using initial states where the AMOC is forced to be in the off state. The 180 ppm and 560 ppm initial states with AMOC *Off* are obtained by
prescribing 0.2 Sv of FWF in the latitudinal belt 50–70°N in the North Atlantic and running the model for 5000 years.

To investigate the stability of the four different AMOC states found from the FWF and $CO_2$ perturbation experiments above in a combined FWF–$CO_2$ space, we interactively designed simulation pathways through this parameter space as shown in

Fig. A2. The rate of change of the forcing in these experiments is again $0.02\,\mathrm{Sv\,kyr^{-1}}$ for FWF and $2\,\%\,\mathrm{kyr^{-1}}$ for $CO_2$. The constant $CO_2$ concentrations used to move in the FWF direction are discretized in steps of $20\,\mathrm{ppm}$ between $180\,\mathrm{ppm}$ and $280\,\mathrm{ppm}$ and in steps of $40\,\mathrm{ppm}$ between $280\,\mathrm{ppm}$ and $560\,\mathrm{ppm}$. The first step was to run experiments with increasing and decreasing FWF, starting from -0.5 Sv and +0.5 Sv, respectively, for all $CO_2$ levels. These simulations are sufficient to trace the stability of the *Off* and *Strong* AMOC states (Fig. A2a,d), because (i) for large positive FWF the AMOC collapses under any $CO_2$ concentration and (ii) for large negative FWF the AMOC always transitions to the *Strong* state. The stability analysis of the *Modern* and *Weak* AMOC states uses the pre-industrial state ($CO_2$ of $280\,\mathrm{ppm}$ and zero FWF) as initial condition, but then requires a more sophisticated procedure to trace their stability through the 2D phase space (Fig. A2b,c).

**Appendix B: Additional figures**

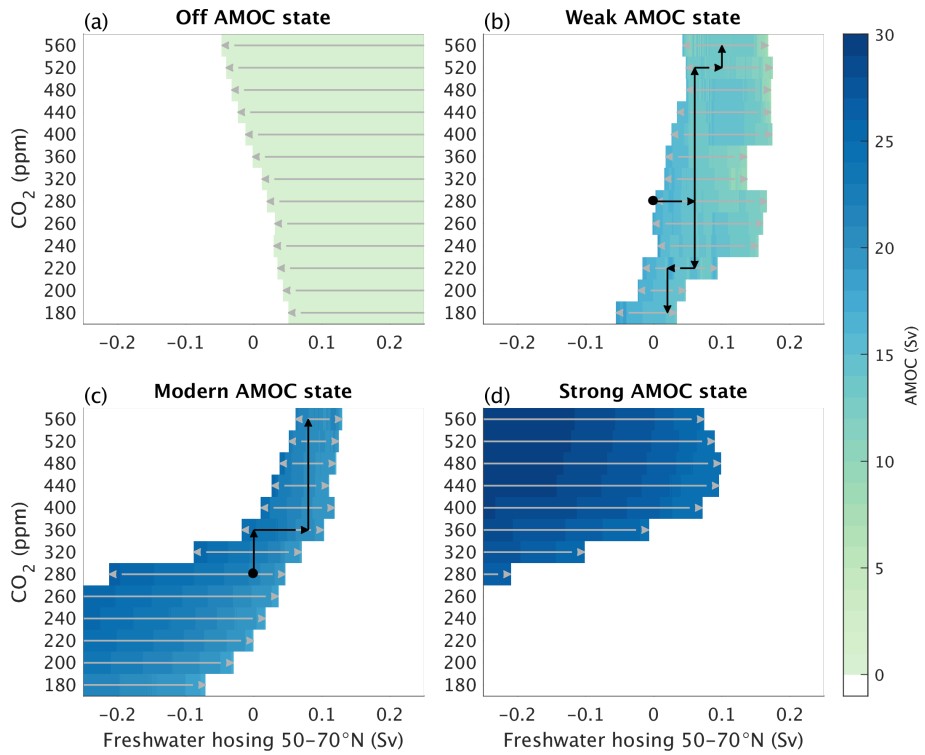

**Figure A2.** Simulation pathways used to explore the stability of the four different AMOC states in the combined $CO_2$ and freshwater space plotted on top of the AMOC stability landscape shown in Fig. 4. The stability of the *Off* AMOC state in (a) was explored with simulations starting from a large FWF of +0.5 Sv and then gradually decreasing the FWF until the AMOC recovers, for all levels of $CO_2$. The stability of the *Strong* AMOC state in (d) was tracked in simulations starting from a large negative FWF of -0.5 Sv and then gradually increasing the FWF. For the investigation of the stability of the (b) *Weak* and (c) *Modern* AMOC states, the starting point were pre-industrial conditions, marked by the black dot. The black arrows indicate the primary path through the $CO_2$ and FWF space, from which subsequent experiments with varying FWF in different directions are initialized (green arrows). Since the *Strong* AMOC state is not stable for $CO_2$ lower than 280 ppm for the FWF range shown in the figure, the stability of the *Modern* AMOC state in (c) for $CO_2$ lower than pre-industrial is diagnosed from simulations initialized with a large negative FWF of -0.5 Sv, similarly to what done in (d) for the *Strong* AMOC state. The rate of change of the forcing in all the experiments is 0.02 Sv kyr$^{-1}$ for FWF and 2 % kyr$^{-1}$ for $CO_2$.

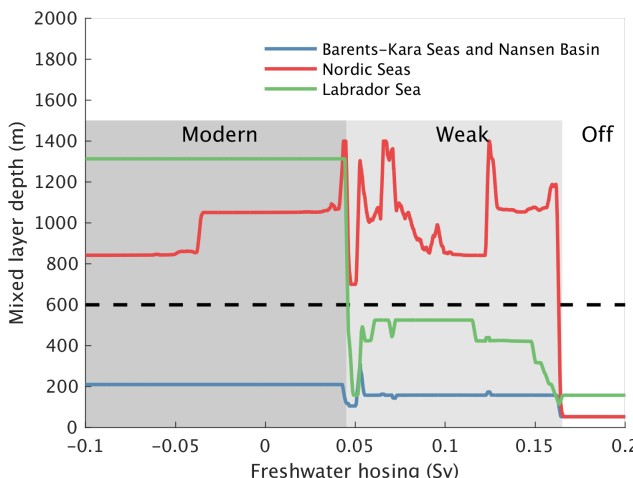

**Figure B1.** Maximum mixed layer depth in the three regions of the North Atlantic that are used to formally categorize the different AMOC states for the experiment with increasing freshwater hosing in the latitudinal belt between 50 and 70°N, corresponding to the solid red curve in Fig. 1. The black dashed line indicates the critical mixed layer depth of 600 m used to discriminate between different convection patterns and AMOC states.

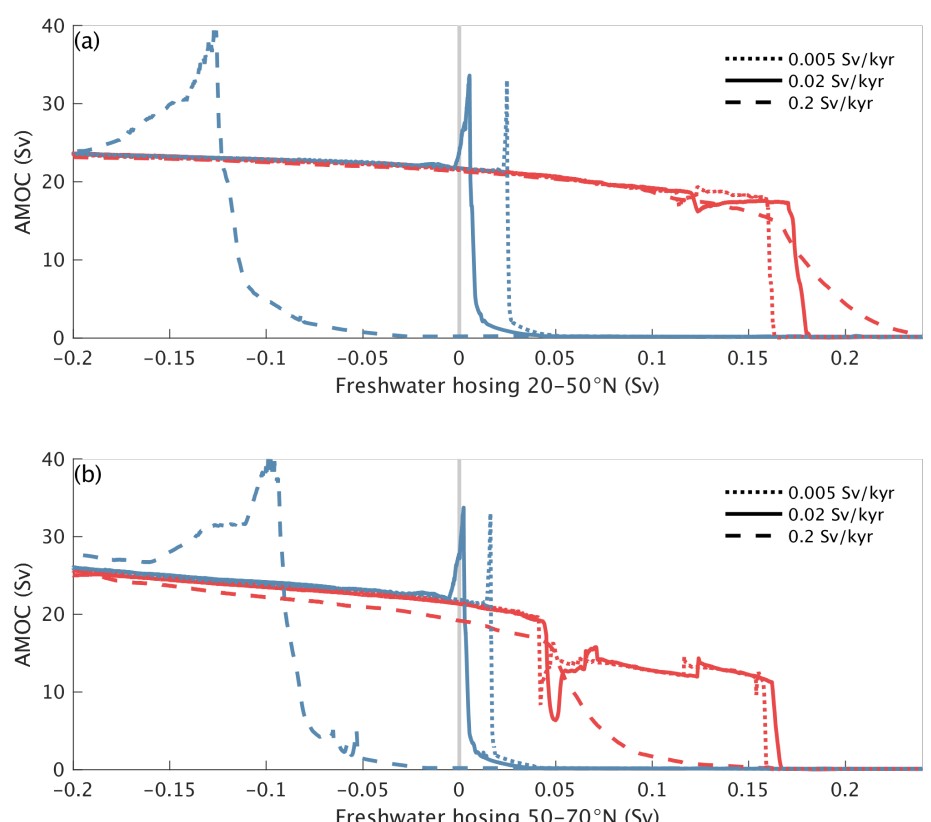

**Figure B2.** Rate-dependence of the hysteresis of the AMOC in freshwater space. AMOC response to prescribed changes in FWF in the latitudinal belts (a) 20–50°N and (b) 50-70°N. The red lines are from simulations with increasing FWF starting at -0.5 Sv and the blue lines are for experiments with decreasing FWF starting at +0.5 Sv. The continuous lines are with the reference rate of change of the imposed FWF of $0.02\,\mathrm{Sv\,kyr^{-1}}$ as also shown in Fig. 1, while the dashed lines represent simulations with a ten-fold increase in the rate of change of hosing ($0.2\,\mathrm{Sv\,kyr^{-1}}$) and the dotted lines are for simulations with an even slower rate of change of $0.005\,\mathrm{Sv\,kyr^{-1}}$.

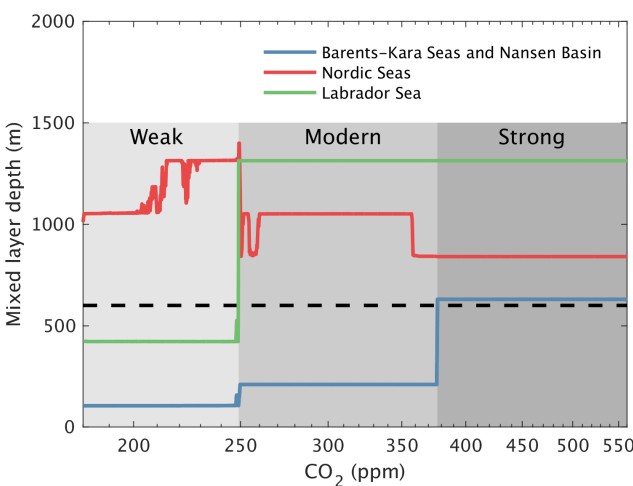

**Figure B3.** Maximum mixed layer depth in the three regions of the North Atlantic that are used to formally categorize the different AMOC states for the experiment with increasing $CO_2$, corresponding to the solid red curve in Fig. 2. The black dashed line indicates the critical mixed layer depth of $600\,\text{m}$ used to discriminate between different convection patterns and AMOC states.

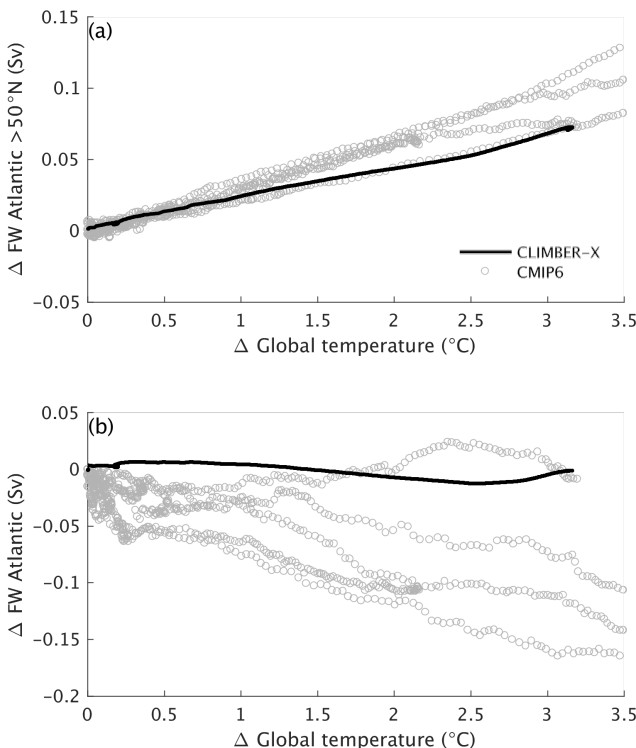

**Figure B4.** Change in the net freshwater flux into the ocean as a function of global temperature change in transient historical and future simulations under the SSP2-4.5 scenario until the year 2300 CE for (a) the northern North Atlantic and Arctic (north of $50°$N) and (b) the whole Atlantic ocean. The solid line is for CLIMBER-X results and the circles represent CMIP6 model results.

*Author contributions.* MW and AG conceived and designed the study. MW performed the model simulations and created the figures. MW and AG wrote the paper.

*Competing interests.* The authors declare that they have no conflict of interest.

*Acknowledgements.* MW is funded by the German climate modeling project PalMod supported by the German Federal Ministry of Education and Research (BMBF) as a Research for Sustainability initiative (FONA) (grant nos. 01LP1920B, 01LP1917D, 01LP2305B). The authors gratefully acknowledge the European Regional Development Fund (ERDF), the German Federal Ministry of Education and Research and the Land Brandenburg for supporting this project by providing resources on the high performance computer system at the Potsdam Institute for Climate Impact Research. We would like to thank Yvan Romé and two anonymous reviewers for their valuable comments, which allowed us

to substantially improve the manuscript.

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
