# Peer review of "Generalized stability landscape of the Atlantic Meridional Overturning Circulation"

_EGUsphere, 2024_

## Author Comment (AC2)

**Reviewer #1**

**General comments**

The manuscript explores the combined effect of changes in CO2 concentrations and North Atlantic freshwater forcing on the existence of multiple AMOC modes in the fast climate model ClimberX. After plotting the AMOC intensity hysteresis cycles resulting from the two independent parameters and including discussions about the role of the initial AMOC state and the rate of change, Willeit and Ganopolski produced a stability landscape of the AMOC modes in the input space formed by the CO2 concentrations and the freshwater forcing.

This work offers a new multidimensional approach to understanding AMOC stability in climate models. The stability landscape map is a convincing and comprehensive way to explore the domain of occurrence of AMOC mode shifts. The volume of simulations produced for this article is remarkable, and the experimental design allows advanced conclusions on the effect of AMOC mode shifts on the climate.

I have, nonetheless, major reservations about the clarity of the manuscript, as well as the justification of some arguments. The manuscript is, at times, difficult to read, and the description of the experiment and the calculations need to be revised to be able to evaluate the validity of the conclusions.

In summary, this paper is a strong contribution to the highly relevant question of AMOC manuscript in climate models. However, some work is required on the text to support the interpretations and conclusions. I recommend major revisions of the manuscript before publication in ESD. The main concerns I identified are the following.

We would like to thank the Reviewer for the detailed review of our paper. The raised comments will help to improve the clarity of the manuscript and are therefore gratefully acknowledged. We will address the Reviewers' comments as outlined below.

- The abstract is not reflective of the work and inconsistent with the conclusion, which is very clear. In particular, the abstract implies that the main aim of the paper is the impact of CO2 concentrations on AMOC stability, instead of, as it is written in the conclusion, performing "a systematic analysis of the AMOC stability in the FWF–CO2 space."

We do not see any contradiction between the fact that in the paper we performed "a systematic analysis of the AMOC stability in the FWF-CO2 space", while in the abstract we focused on the CO2 dimension of this space. This is because the freshwater dimension of this phase space has already been studied in many previous studies, including ours. However, to focus the abstract a bit more on the AMOC stability landscape instead of the CO2 response alone, we will add the following:
*Generally, strong AMOC states are favored by high CO2 concentrations and negative FWF, while weak AMOC states are characteristic for CO2 levels below pre-industrial or positive FWF. Our AMOC stability landscape helps to explain AMOC instability in colder climates and provides useful context to interpret possible future AMOC trajectories in a warmer climate. In the absence of external FWF, e.g. from melting ice sheets, the model shows an increase in equilibrium AMOC strength with increasing CO2 levels.*

- In the introduction, the impact of CO2 on the AMOC stability is said to "remain largely unexplored", and the freshwater forcing outside of the 20-50°N band to be a rare occurrence. I believe both of these views are outdated, and the introduction is missing key references and discussion points that provide an accurate and comprehensive picture of the current state of the research. If these comments only concerned modern days, it should clearly be stated, and the results from the palaeo community should be discussed. I recommend splitting the second paragraph of the introduction into three parts. A first one on the CO2 effect, including missing references (e.g. Brown and Galbraith 2016, Zhang 2017, Klockmann 2018, Vettoretti 2022), a second one on the FWF including missing references (e.g. Smith and Gregory 2009, Roche 2010, Kageyama 2013, Ivanovic 2018, Romé 2022) and a third one on the need for combined CO2xFWF analysis and an introduction of your paper, which is currently too short.

We agree that the statement "remain largely unexplored" is indeed too strong. Some studies have already explored the AMOC response to CO2, although they largely concentrate on the paleo-context. We will discuss that in the revised paper. We have already cited Brown and Galbraith (2016) and Klockmann et al. (2018) in our paper. At the same time, Zhang et al. (2017), Vettoretti et al. (2022) and the reviewer's own paper (Romé et al. 2022) are cited in another paper of ours which is devoted to Dansgaard-Oeschger events (Willeit et al., 2024, Climate of the Past Discussion). However, we think that these papers are not relevant to the present manuscript.

The cited FWF hysteresis experiments have all been performed with freshwater hosing between 20-50°N. We believe that transient water hosing experiments, that indeed focused also on different areas of the north Atlantic, are not relevant in the context of our paper.

The same applies to the second group of references suggested by the reviewer (Smith and Gregory 2009, etc...). They are all about transient water hosing experiments. We do not believe that there is a need to explain the fundamental difference between tracing the AMOC stability diagram and performing transient water hosing experiments. The number of papers describing such experiments, starting with the classical Manabe and Stouffer (1995) paper, is enormous and we cannot see any benefit for our paper and potential readers in citing these publications.

Altogether, we will rewrite the second part of the introduction with the following outline:

1. FWF hysteresis
2. CO2 effect
3. Combined CO2xFWF and introduction of our paper

as follows:

*Stocker and Wright (1991) and Rahmstorf (1995) pioneered the use of surface freshwater forcing (FWF) experiments to analyze the stability of the AMOC and showed a hysteresis behavior in ocean models. Since then, models of different complexity have found that the AMOC shows a hysteresis behavior to FWF that is associated with multiple stable states (Ganopolski and Rahmstorf, 2001; Rahmstorf et al., 2005; Hawkins et al., 2011; Hu et al., 2012; van Westen and Dijkstra, 2023; Ando and Oka, 2021; Hofmann and Rahmstorf, 2009; Gregory et al., 2003; Lenton et al., 2009), although there is no consensus as to whether the AMOC is in a monostable or a bistable regime under present climate conditions (e.g. Weijer et al., 2019; Liu et al., 2017). While most of these hysteresis experiments have been performed under pre-industrial or present-day conditions, some have considered the*

*dependence on background climate by exploring the hysteresis behavior also for the last glacial maximum (Ganopolski and Rahmstorf, 2001; Schmittner et al., 2002; Ando and Oka, 2021; Pöppelmeier et al., 2021; Weber and Drijthout, 2007; Prange et al., 2002). Most of the hysteresis experiments have been performed with FWF in the latitudinal belt between 20–50°N in the Atlantic, thereby avoiding a direct perturbation of the convection sites further north in order to focus on the salt-advection feedback. FWF applied in the convection areas has a stronger impact on the AMOC (e.g. Ganopolski and Rahmstorf, 2001; Smith and Gregory, 2009), because the state of the AMOC is tightly linked to the production of deep water.*

*Convection and deep water formation do not only depend on surface freshwater flux, but are more generally controlled by the surface buoyancy flux, which also depends on the sensible heat cooling and the temperature at the sea surface. The temperature dependence of the surface buoyancy flux arises from the nonlinear equation of state of seawater, and in particular from the temperature dependence of the thermal expansion coefficient (Roquet et al., 2015). Perturbations to the climate will affect both the net surface freshwater flux, as a result of changes in the hydrological cycle, and the surface temperature, with intricate implications for AMOC stability. The effect of climate on AMOC stability has been investigated in relatively few studies, mainly by changing the concentration of atmospheric $CO_2$ (Brown and Galbraith, 2016; Klockmann et al., 2018; Galbraith and de Lavergne, 2019). These studies show a generally stronger AMOC in equilibrium with higher $CO_2$, but mostly focused on climates colder than present. Recently, Gérard and Crucifix (2024) have performed model simulations with slowly increasing and decreasing $CO_2$, producing an AMOC hysteresis in $CO_2$ space and suggesting an AMOC weakening in equilibrium with a warmer climate.*

*For an improved understanding of past and future AMOC evolution it is important to consider changes in climate and changes in the surface ocean freshwater balance due to changing land ice volume, since both are expected to play an important role for AMOC stability. Here we use an Earth system model to systematically explore the combined effect of surface FWF and climate on AMOC stability. The effect of external FWF is quantified by running experiments with FWF in different latitudinal belts in the North Atlantic, while the effect of climate is explored by varying the atmospheric $CO_2$ concentration, which is one of the main factors driving past and future climate changes.*

- I do not believe that the authors can claim to be the first to attempt to draw a landscape of AMOC stability in the $CO_2$ x freshwater forcing space is true, see Brown and Galbraith 2016 for instance. However, I would say that this paper presents the most comprehensive and robust method up to date. If this claim only applies to modern-day studies of the AMOC, it needs to be clearly stated and put in context with palaeo studies.

To our knowledge we are the first who "have performed a systematic analysis of the AMOC stability in the FWF–$CO_2$ space". Just to clarify the terminology: under stability (diagram) or phase portrait of the AMOC, we understand the 2D extension of the classical 1D Stommel's hysteresis diagram, which depicts the strength of different possible, stable AMOC modes and the positions of bifurcation transitions between different modes. All previous studies explore either one or the other dimension but not the two together. As far as the Brown and Galbraith (2016) paper is concerned, it does not aim to produce a stability diagram of AMOC. They

performed a set of quasi-equilibrium simulations of the climate for different combinations of CO2, ice sheets and orbital parameters, as well as several transient water hosing experiments.

- Significant mode shifts and overshoots on the hysteresis cycles are not discussed in the text. In particular, in Figure 1, the transition in the red solid line around 0.05Sv is remarkable: Is it different from an overshoot? Why is it sustained for about 1000 years? Could this be an occurrence of millennial-scale variability? Could you link this to Willeit 2024?

We will add the following text to clarify this:
*This abrupt AMOC weakening is linked to the DO-variability produced by the model in the presence of noise, as shown in Willeit et al. (2024), and the associated overshoot in Fig. 1 is actually attributable to a missed initiation of millennial-scale oscillations.*

- The definition of the different states comes too late in the paper and lacks precision. How do you define the different modes, using the AMOC index, the mixed layer depth or manually? Additionally, How do you calculate the AMOC index? What references did you use for the modern-day deep water formation sites, and how do they compare to your modern mode? Over what time slices was Figure 3 plotted?

The different modes are defined based on the mixed layer depth in different regions in the North Atlantic, as detailed below. This will be explicitly added to the caption of Fig. 4.
The modern-day is characterized by deep water formation in the Labrador Sea and the Nordic Seas, which matches the locations with a deep mixed layer in the Modern AMOC state. Similarly to Fig. 6, also the AMOC states in Fig. 3 are for the boundary conditions of 400 ppm and 0.05 Sv, where all 4 AMOC states co-exist in the model. This will be added to the caption for clarification.

- The interpretation of the freshwater flux needs to be clarified in this article, and it becomes a problem when comparing pre-industrial to modern conditions. Would it not be more accurate to account for changes in CO2 and freshwater forcing when comparing the two? Otherwise, what is the point of using a two-dimensional landscape? In addition, the following statement from the conclusion "Our results indicate a generally stronger and deeper AMOC at equilibrium under warmer climate conditions. This is in contrast to the projected AMOC weakening response to anthropogenic global warming […]" is only valid if one considers the sole CO2 effect, but freshwater forcing is expected to increase with Greenland melt, which could take us into a region of the landscape where all four modes exist. I think the comparison between past, present and future states should include a discussion about the role of excess freshwater induced by ice sheet melting.

Our stability diagram shows the equilibrium states of the climate (atmosphere-ocean-sea ice-land) system with fixed present-day ice sheets. Past or future freshwater flux from the surrounding ice sheets is considered as the external forcing and thus is the second dimension. We will add the following text to the 'AMOC hysteresis in freshwater space' section to clarify this point:
*The FWF, as used in this study, represents perturbations to the freshwater balance of the North Atlantic by factors external to the climate (atmosphere-ocean-sea ice-land) system, namely from changing land ice volume that is not accounted for in our simulations because we use prescribed present-day ice sheets.*

Given the strongly transient nature of ongoing global warming it does not make sense to compare pre-industrial with present-day conditions in the context of the AMOC stability landscape. As already mentioned in the paper, the transient and equilibrium AMOC response to $CO_2$ are fundamentally different, with an increase in AMOC strength with increasing $CO_2$ under equilibrium conditions, but a weakening if the $CO_2$ increase is fast. The non-trivial relation between future projected AMOC evolution and the stability landscape will be investigated in a forthcoming paper. Also, to avoid possible confusion on the interpretation of the red dot in Fig. 4, we will remove it. It was originally simply intended to show the present-day $CO_2$ concentrations, but it could be misinterpreted in terms of the present-day AMOC state.

During glacial terminations, the anomalous freshwater flux into the North Atlantic exceeded 0.1 Sv and had a profound impact on the AMOC, while for present-day conditions and even for the whole considered range of $CO_2$ (up to 2x$CO_2$), the net freshwater flux from Greenland is small (<< 0.1 Sv (Otosaka et al., 2023; Briner et al., 2020; Calov et al., 2018)) and has little effect on the AMOC. We will add this information to the revised manuscript. For higher $CO_2$, the future freshwater flux from Greenland can be significant, but for high $CO_2$, Greenland will melt completely in around thousand years, which is much shorter than the time needed to trace the stability diagram. This highlights again that the only sensible way to deal with net freshwater from ice sheets when tracing the AMOC stability diagram is to treat it as external forcing.

Similarly to what done in the abstract, also in the conclusions we will make clear that the statement '*Our results indicate a generally stronger and deeper AMOC at equilibrium under warmer climate conditions…*' is strictly valid only if meltwater input from Greenland is small enough, which is clearly the case according to different studies cited above.

- The details about the construction of the stability landscape is lacking precision and its validity cannot be evaluated. This all the more important as you highlighted the dependence of the direction of variation in Figure 1,2 and B1.

We will add detailed information on how the stability landscape is constructed in the caption of Fig. A1:
*Simulation pathways used to explore the stability of the four different AMOC states in the combined $CO_2$ and freshwater space plotted on top of the AMOC stability landscape shown in Fig. 4. The stability of the Off AMOC state in (a) was explored with simulations starting from a large FWF of +0.5 Sv and then gradually decreasing the FWF until the AMOC recovers, for all levels of $CO_2$. The stability of the Strong AMOC state in (d) was tracked in simulations starting from a large negative FWF of -0.5 Sv and then gradually increasing the FWF. For the investigation of the stability of the (b) Weak and (c) Modern AMOC states, the starting point were pre-industrial conditions, marked by the black dot. The black arrows indicate the primary path through the $CO_2$ and FWF space, from which subsequent experiments with varying FWF in different directions are initialized (green arrows). Since the Strong AMOC state is not stable for $CO_2$ lower than 280 ppm for the FWF range shown in the figure, the stability of the Modern AMOC state in (c) for $CO_2$ lower than pre-industrial is diagnosed from simulations initialized with a large negative FWF of -0.5 Sv, similarly to what done in (d) for the Strong AMOC state. The rate of change of the forcing in all the experiments is 0.02 Sv/kyr for FWF and 2 %/kyr for $CO_2$.*

**Specific comments**

*Abstract*

- L8 - Can you briefly define the OFF and Modern AMOC states?

We will add the following brief description:
*'Apart from an Off AMOC state with no North Atlantic deep water formation and a Modern-like AMOC with deep water forming in the Labrador and Nordic Seas as observed at present,...'*

- L11-12 ("In general, the model shows an increase in equilibrium AMOC strength for higher CO2 levels.") - This does not reflect the actual nature of the work, which goes way beyond this sole aspect. This statement is valid for the standard CO2 experiment in Figure 2, but not consistent with Figure 4 (ex. in Figure 4a, an increase of CO2 can trigger a weak mode). The abstract needs not to focus only on the CO2 experiment but also on the stability landscape.

See response to first general comment above.

**Introduction**

- L28 ("There is no consensus as to whether the AMOC is in a monostable or a bistable regime under present climate conditions") - This needs a reference; I am aware of discussions on the potential weakening of the AMOC, less so about the current state of the AMOC.

We will add a reference to the review paper by Weijer et al. (2019) and to Liu et al. (2017), who suggested that the AMOC is bistable at present based on observational constraints.

*Results*

- L59 ("In particular, there is a range of FWF over which the AMOC has two stable states has two stable states") – Can you be precise about the range of FWF you are talking about? It also depends on your definition of stability, as I would argue that the dip around 0.05 Sv in Figure 1 is a sign of instability.

We will specify that we are talking about: *'roughly between 0.01 and 0.17-0.18 Sv'*. Yes, the dip around 0.05 Sv marks the transition between the Modern and Weak AMOC states, but doesn't change the fact that in terms of hysteresis curve the AMOC has two stable states for any given FWF in the above-mentioned range.

- L61 - Does "preindustrial conditions" mean 0 Sv in this case? Also, according to the methods, this experiment has pre-industrial CO2 concentrations but a modern-day ice sheet. I would be more careful about using "pre-idustrial" conditions, what about "initial state" instead?

We will specify that pre-industrial conditions refer to 280 ppm of $CO_2$. We consider pre-industrial and present-day ice sheets to be the same, which is a very reasonable assumption.

- L63 ("suggesting a prominent role of convective instability") - Could you show that this is a convective instability, showing deep water formation sites activity, for example?

The potential energy released by convective mixing as a function of time in the simulations shown in Fig. 1 in the paper is shown in Fig. 1 below, separately for the Nordic Seas and the Labrador Sea. The figure clearly shows the abrupt decrease of convective activity associated with the AMOC transitions, providing a clear indication of convective instability occurring in the model.

[Figure]

*Figure 1 Potential energy released by convection in the Nordic Seas (above) and the Labrador Sea (below) as a function of time for the FWF hysteresis simulations with increasing FWF shown in Fig. 1 in the paper.*

- L65 - "This is the result of a collapse of deepwater formation […] of observed past Dansgaard-Oeschger events." : Here again, the manuscript is missing a plot with the deep water sites dynamics to verify this statement, and a reference about convection in D-O records.

We are not aware of any direct reconstructions of convection locations during DO events. We will therefore remove the 'observed' in the sentence and add a reference to Willeit et al (2024), which includes a figure with the simulated mixed layer depth during DO Stadials and Interstadials.

- L67 – A definition of what the authors mean by Off, Weak, Modern-day, Strong is needed at this point of the paper.

We believe that at this point in the paper it is ok to simply refer qualitatively to the different AMOC states shown in Fig. 1, while a strict definition is given later, after all the different states have been gradually introduced.

- L82 ("which is possibly more relevant for the ongoing global warming") - I disagree, both CO2 and meltwater discharge are relevant to future climate changes.

We disagree and in the paper we have explained in detail why we think that the meltwater discharge from Greenland is small compared to the changes in the freshwater balance of the northern North Atlantic induced by changes in the hydrological cycle due to higher CO2 (lines 133-137):

*For CO2 doubling the net freshwater flux into the northern North Atlantic increases by ~0.07 Sv, an amount which is sufficient to cause a transition of the AMOC into a weak state if CO2 is kept constant at pre-industrial values. 0.07 Sv is a relatively large freshwater flux, which is approximately **an order of magnitude higher than the net freshwater flux from the Greenland ice sheet simulated under similar temperatures** (e.g. Calov et al. (2018) and Briner et al. (2020)), and would roughly correspond to the rate of freshwater input resulting from the Greenland ice sheet melting completely over a time period of ~1500 years.*

- L85-93 - I find the wording of this section confusing. Is the rate of CO2 increase in this paper slower than the "slow" increase in Gérard and Crucifix? How do you explain that you see a strengthening of the AMOC when Gérard and Crucifix 2024 saw a decrease? Could this simply mean that the AMOC response to CO2 is highly uncertain and model dependent?

Considering also the comment by Gérard and Crucifix on our pre-print (https://doi.org/10.5194/egusphere-2024-1482-CC1), we have rewritten this paragraph:
*Gérard and Crucifix (2024) recently analyzed the AMOC response to a slow CO2 increase and found a gradual AMOC weakening and eventual collapse at CO2 above ~1500 ppm. This is in contrast to our results, which show an increase in AMOC strength with increasing CO2, at least up to a CO2 concentration of 560 ppm. It should be noted that both in Gérard and Crucifix (2024) and in our study the CO2 increase is slow enough to track the equilibrium AMOC response, but it has been shown that the AMOC response is highly sensitive to the applied rate of temperature change (Stocker and Schmittner, 1997).*

- L105 ("For CO2 above ~250 ppm, the convection pattern resembles the present-day state with deep water forming in the Labrador Sea and in the Nordic Seas") - Are you talking about Willeit et al. 2024 or this manuscript?

We are talking about this manuscript, which should be clear form the fact that later in the same sentence we are explicitly referring to Fig. 3c, which shows the mixed layer depth for the *modern* AMOC state.

- L156 ("If the climate would be in equilibrium with present-day CO2 concentrations of ~420 ppm, the model suggests that the Modern AMOC state would not be stable, but that the AMOC would rather be in the Strong state instead") – Back to the point about the interpretation of future freshwater forcing, would the accelerated melt of Greenland not move the system along the Modern AMOC conditions diagonal instead? Otherwise, this is a major caveat of the analysis that needs to be discussed

(although, arguably, the last paragraph of the discussion introduces this idea of model dependency).

Yes, the melt of Greenland would move the system in diagonal direction, but because freshwater input from Greenland melt is relatively small for the considered levels of global warming, the system would still be quite close to the FWF=0 axis. Also, as discussed above, this paper is really about the AMOC stability diagram and its relation to transient future climate and AMOC change will be the subject of further investigations.

- **Figure 4** - Good Figure but missing information about the criteria used for the clustering of the AMOC states. Is it purely based on the AMOC index, or on the mixed layer depth?

We will add the following text in the figure caption to clarify this:
*The different states are formally defined based on a critical threshold ($mld_{max}^{crit}$=600 m) of the maximum mixed layer depth ($mld_{max}$) in three separate regions in the North Atlantic, namely (i) the Nordic Seas, (ii) the Labrador Sea and (iii) the Barents and Kara Seas and the Nansen basin.*
*Off : $mld_{max}$< $mld_{max}^{crit}$ in (i-iii);*
*Weak: $mld_{max}$> $mld_{max}^{crit}$ in (i) and $mld_{max}$< $mld_{max}^{crit}$ in (ii-iii);*
*Modern: $mld_{max}$> $mld_{max}^{crit}$ in (i-ii) and $mld_{max}$< $mld_{max}^{crit}$ in (iii);*
*Strong: $mld_{max}$> $mld_{max}^{crit}$ in (i-iii).*

- **Figure 5** - I think this is the most important Figure of the paper and it deserves to be bigger. Also, why are the Off mode treated differently than the different modes? It makes the Figure difficult to interpret at first reading. Could you use different colours or shading and include all four states in the stability landscape?

We agree that this is the most important figure in the paper and will consequently make it larger. The Off state is in some sense fundamentally different from any of the 'on' states and it can not easily be framed in the general trend of AMOC becoming weaker with decreasing CO2 and increasing FWF, which is what the color shading is intended to represent. Representing the Off state with the same color scheme would be problematic when e.g. considering the region of the landscape where the Strong and Off state are stable. What color would be associated to the coexistence of these two extreme states?
We will instead replace the crosses, which indicate the stability region of the Off state, with a white hatched area. White will therefore represent the Off state in the new figure, as shown in Fig. 2 below. This will hopefully make the figure easier to read and interpret. We will also add the following text to the figure caption:
*The filled white area indicates where only the Off AMOC state is stable, while the white hatched area shows the domain where the Off AMOC state and one or more of the three 'on' AMOC states coexist.*

[Figure]

*Figure 2 Revised Fig. 5.*

- L181 - This is a good section and a convincing way to perform this analysis. However, as you showed, the sea ice extent is driving the most significant temperature changes, so I think it would be good to conclude on the impact of less extensive Arctic sea ice in future climate on these results.

We will add the following sentence to clarify this point:
*While the temperature differences in Fig. 6 are representative of the impact of the different AMOC states on climate, they are strictly valid only for an atmospheric CO2 of 400 ppm and a freshwater forcing of 0.05 Sv and could differ if the transition between AMOC states occurs under different boundary conditions.*

***Discussion***

- L186 - Can you say a word about the Modern and Off AMOC modes.

We will add some more information on these two states as follows:
*Apart from an Off AMOC state with no North Atlantic deep water formation and a Modern-like AMOC state with deep water forming in the Labrador and Nordic Seas as observed at present,…*

***SI***

- L256 - The paths used in **Figure A1** need to be clearly explained in the Figure or the text, otherwise it is impossible to validate the protocol in Figure 4. Do they

correspond to the black or the green arrows? Do they include the standard experiments?

We will explain in detail the paths in Fig. A1 in an expanded caption:
*Simulation pathways used to explore the stability of the four different AMOC states in the combined CO2 and freshwater space plotted on top of the AMOC stability landscape shown in Fig. 4. The stability of the Off AMOC state in (a) was explored with simulations starting from a large FWF of +0.5 Sv and then gradually decreasing the FWF until the AMOC recovers, for all levels of CO2. The stability of the Strong AMOC state in (d) was tracked in simulations starting from a large negative FWF of -0.5 Sv and then gradually increasing the FWF. For the investigation of the stability of the (b) Weak and (c) Modern AMOC states, the starting point were pre-industrial conditions, marked by the black dot. The black arrows indicate the primary path through the CO2 and FWF space, from which subsequent experiments with varying FWF in different directions are initialized (green arrows). Since the Strong AMOC state is not stable for CO2 lower than 280 ppm for the FWF range shown in the figure, the stability of the Modern AMOC state in (c) for CO2 lower than pre-industrial is diagnosed from simulations initialized with a large negative FWF of -0.5 Sv, similarly to what done in (d) for the Strong AMOC state. The rate of change of the forcing in all the experiments is 0.02 Sv/kyr for FWF and 2 %/kyr for CO2.*

- **Figure B2** does not have a caption.

We apologize for having overlooked the missing caption and will add the following caption to Fig. B2:
*Change in the net freshwater flux into the ocean as a function of global temperature change in transient historical and future simulations under the SSP2-4.5 scenario until the year 2300 CE for (a) the northern North Atlantic and Arctic (north of 50° N) and (b) the whole Atlantic ocean. The solid line is for CLIMBER-X results and the circles represent CMIP6 model results.*

**Technical corrections**

***Introduction***

- L17 - Do you have an example of a "societal" change?

We will remove 'societal' as we focus only on the climate impact.

- L19 - Add Bellomo 2021 for state-of-the-art climate model references.

We will add a references to Bellomo et al. (2021), Weijer et al. (2020), Weaver et al. (2012) and Romanou et al. (2023).

***Results***

- L51 – "Willeit et al. (2020)" to "(Willeit et al., 2020)

Will be fixed, thanks.

- L69 ("most AMOC hysteresis experiments to date have been performed with FWF at lower latitudes (usually between 20N and 50N)") – Needs references.

We will add a reference to Rahmstorf et al. (2005), Hu et al. (2012) and van Westen and Dijkstra (2023).

- L71 ("In our hysteresis experiment") – Change to "In Figure 1", there are multiple hysteresis experiments in this paper.

We will change this to 'In our freshwater hysteresis experiment in Fig. 1'.

- L80 ("give the wrong impression that the AMOC Off state is also stable") – Is it a wrong impression or relative to the rate of meltwater discharge? Can you give an order of magnitude of the expected meltwater discharge for future melting or during a D-O event/Heinrich event?

The idea of the hysteresis experiments is to find the equilibrium states of the AMOC. To do this, the rate of change of freshwater forcing should be small enough to actually track the equilibrium states. The rate of change in our standard experiments is low enough to have confidence that the produced stability diagram is accurate enough. Obviously, the rate of change of 0.2 Sv/kyr corresponding to the dashed lines in Fig. B1 is not low enough and therefore the hysteresis curve produced with such rate of changes is wrong. These are idealized experiments that have little to do with past or future rates of meltwater discharge.

- L83 – "In an experiment where" to "In Figure 2"?

Will be changed as suggested.

- L94 ("two discrete transition") - what do you mean by discrete? Abrupt?

We will replace discrete with abrupt.

- **Figure 2** - It is difficult to distinguish the solid and dotted lines on this Figure. Maybe two columns?

We will make the continuous lines in (a) slightly transparent, so that the dashed lines will be visible on top.

- L110 ("but this has been shown to not be a requirement for the existence of millennial-scale variability") – Vague, can you say more?

We will expand the discussion of this point as follows:
*A narrow window of $CO_2$ concentrations exists for which both convection patterns are stable for the same $CO_2$, but in Willeit et al. (2024) it was shown that this bistability is not a requirement for the existence of millennial-scale variability, for which it is sufficient that the system is close to where the abrupt AMOC transition between two different states occurs in the phase space of $CO_2$.*

- **Figure 3** - The Figure needs to be bigger. Please add labels to the colour-bars. Does seasonal mean winter in this case?

We will either make the figure bigger, if possible for ESD standards, or alternatively switch rows and columns.
What is meant with 'seasonal maximum' is the monthly mean maximum over the year. We will change this to 'maximum monthly mean…of the year' the revised paper.

- L130 - "net freshwater flux" to "net surface freshwater flux".

Will be changed.

- L133 ("for CO2 doubling") – I would say "for double the amount of CO2" because you are not doing a CO2 doubling experiment, which, as you highlighted, could have a different impact on the AMOC.

Good point, thanks. Will be changed as suggested.

- L139 - "in the model the net surface freshwater flux into the whole Atlantic Ocean shows the opposite trend" to "the net surface freshwater flux into the whole Atlantic Ocean shows the opposite trend in our model"?

Will be changed.

- L144 - remove "a stabilizing effect"?

Will be removed.

- L150 - "can be investigating by tracing their stability through" to "can be investigated by tracing the AMOC response in"

Will be changed.

- L169 - "to explore the pure effect of the different AMOC states on climate" to "to isolate the effect of the changes of AMOC states on the climate".

Will be changed.

- **Figure 6**: Can you use a different colour scheme for the temperature?

As suggested also by reviewer #3, we will change the color scale to make the figure easier to read.

***Discussion***

- L193 - "explains" to "demonstrates"?

Will be changed to avoid repetition in the use of 'explains'.

- L202 ("anthropogenic global warming") - Add Bellomo 2021 to the references for state-of-the-art climate models.

We will add a citation to Bellomo 2021.

*SI*

- **Figure B1**: Hard to distinguish the solid and dotted lines on this Figure. Maybe two columns?

We think that it is useful to see all three curves in the same figure to have a direct comparison between the three rates and the different curves seem to be well distinguishable where they don't overlap, which is what we are interested in.

**References**

Bellomo, K., Angeloni, M., Corti, S., and von Hardenberg, J.: Future climate change shaped by inter-model differences in Atlantic meridional overturning circulation response, Nat. Commun., 12, 1–10, https://doi.org/10.1038/s41467-021-24015-w, 2021.

[revised manuscript text omitted]

---

## Author Response (AR1)

**Reviewer #1**

**General comments**

The manuscript explores the combined effect of changes in CO2 concentrations and North Atlantic freshwater forcing on the existence of multiple AMOC modes in the fast climate model ClimberX. After plotting the AMOC intensity hysteresis cycles resulting from the two independent parameters and including discussions about the role of the initial AMOC state and the rate of change, Willeit and Ganopolski produced a stability landscape of the AMOC modes in the input space formed by the CO2 concentrations and the freshwater forcing.

This work offers a new multidimensional approach to understanding AMOC stability in climate models. The stability landscape map is a convincing and comprehensive way to explore the domain of occurrence of AMOC mode shifts. The volume of simulations produced for this article is remarkable, and the experimental design allows advanced conclusions on the effect of AMOC mode shifts on the climate.

I have, nonetheless, major reservations about the clarity of the manuscript, as well as the justification of some arguments. The manuscript is, at times, difficult to read, and the description of the experiment and the calculations need to be revised to be able to evaluate the validity of the conclusions.

In summary, this paper is a strong contribution to the highly relevant question of AMOC manuscript in climate models. However, some work is required on the text to support the interpretations and conclusions. I recommend major revisions of the manuscript before publication in ESD. The main concerns I identified are the following.

*We would like to thank the Reviewer for the detailed review of our paper. The raised comments helped to improve the clarity of the manuscript and are therefore gratefully acknowledged. We have addressed the Reviewers' comments as outlined below.*

- The abstract is not reflective of the work and inconsistent with the conclusion, which is very clear. In particular, the abstract implies that the main aim of the paper is the impact of CO2 concentrations on AMOC stability, instead of, as it is written in the conclusion, performing "a systematic analysis of the AMOC stability in the FWF–CO2 space."

*We do not see any contradiction between the fact that in the paper we performed "a systematic analysis of the AMOC stability in the FWF-CO2 space", while in the abstract we focused on the CO2 dimension of this space. This is because the freshwater dimension of this phase space has already been studied in many previous studies, including ours. However, to focus the abstract a bit more on the AMOC stability landscape instead of the CO2 response alone, we have modified the second part of the abstract as follows:*
*"The Off and Weak states are stable for the entire range of CO2, but only for positive FWF. The Modern state is stable under higher than preindustrial CO2 for a range of positive FWF and for lower CO2 only for negative FWF. Finally, the Strong state is stable only for CO2 above 280 ppm and FWF<0.1 Sv. Generally, the strength of the AMOC increases with increasing CO2 and decreases with increasing FWF.*
*Our AMOC stability landscape helps to explain AMOC instability in colder climates and, although it is not directly applicable to the fundamentally transient AMOC response to global*

*warming on a centennial time scale, it can provide useful information about the possible long-term fate of the AMOC. For instance, while under preindustrial conditions the AMOC is monostable in the model, the Off state also becomes stable for CO2 concentrations above ~400 ppm, suggesting that an AMOC shutdown in a warmer climate might be irreversible."*

This will also help to interpret our results in the context of past and future climate change and highlight the clear distinction that has to be made between transient and equilibrium response.

- In the introduction, the impact of CO2 on the AMOC stability is said to "remain largely unexplored", and the freshwater forcing outside of the 20-50°N band to be a rare occurrence. I believe both of these views are outdated, and the introduction is missing key references and discussion points that provide an accurate and comprehensive picture of the current state of the research. If these comments only concerned modern days, it should clearly be stated, and the results from the palaeo community should be discussed. I recommend splitting the second paragraph of the introduction into three parts. A first one on the CO2 effect, including missing references (e.g. Brown and Galbraith 2016, Zhang 2017, Klockmann 2018, Vettoretti 2022), a second one on the FWF including missing references (e.g. Smith and Gregory 2009, Roche 2010, Kageyama 2013, Ivanovic 2018, Romé 2022) and a third one on the need for combined CO2xFWF analysis and an introduction of your paper, which is currently too short.

We agree that the statement "remain largely unexplored" is indeed too strong. Some studies have already explored the AMOC response to CO2, although they largely concentrate on the paleo-context. We discuss that in the revised paper. We have already cited Brown and Galbraith (2016) and Klockmann et al. (2018) in our paper. At the same time, Zhang et al. (2017), Vettoretti et al. (2022) and the reviewer's own paper (Romé et al. 2022) are cited in another paper of ours which is devoted to Dansgaard-Oeschger events (Willeit et al., 2024, Climate of the Past Discussion). However, we think that these papers are not relevant to the present manuscript.
The cited FWF hysteresis experiments have all been performed with freshwater hosing between 20-50°N. We believe that transient water hosing experiments, that indeed focused also on different areas of the north Atlantic, are not relevant in the context of our paper.
The same applies to the second group of references suggested by the reviewer (Smith and Gregory 2009, etc...). They are all about transient water hosing experiments. We do not believe that there is a need to explain the fundamental difference between tracing the AMOC stability diagram and performing transient water hosing experiments. The number of papers describing such experiments, starting with the classical Manabe and Stouffer (1995) paper, is enormous and we cannot see any benefit for our paper and potential readers in citing these publications.
Altogether, we have rewritten the second part of the introduction with the following outline:

1. FWF hysteresis
2. CO2 effect
3. Combined CO2xFWF and introduction of our paper

as follows:

*Stocker and Wright (1991) and Rahmstorf (1995) pioneered the use of surface freshwater forcing (FWF) experiments to analyze the stability of the AMOC and showed a hysteresis*

*behavior in ocean models. Since then, models of different complexity have found that the AMOC shows a hysteresis behavior to FWF that is associated with multiple stable states* (Ganopolski and Rahmstorf, 2001; Rahmstorf et al., 2005; Hawkins et al., 2011; Hu et al., 2012; van Westen and Dijkstra, 2023; Ando and Oka, 2021; Hofmann and Rahmstorf, 2009; Gregory et al., 2003; Lenton et al., 2009), *although there is no consensus as to whether the AMOC is in a monostable or a bistable regime under present climate conditions (e.g.* Weijer et al., 2019; Liu et al., 2017). *While most of these hysteresis experiments have been performed under pre-industrial or present-day conditions, some have considered the dependence on background climate by exploring the hysteresis behavior also for the last glacial maximum* (Ganopolski and Rahmstorf, 2001; Schmittner et al., 2002; Ando and Oka, 2021; Pöppelmeier et al., 2021; Weber and Drijthout, 2007; Prange et al., 2002). *Most of the hysteresis experiments have been performed with FWF in the latitudinal belt between 20– 50°N in the Atlantic, thereby avoiding a direct perturbation of the convection sites further north in order to focus on the salt-advection feedback. FWF applied in the convection areas has a stronger impact on the AMOC (e.g.* Ganopolski and Rahmstorf, 2001; Smith and Gregory, 2009), *because the state of the AMOC is tightly linked to the production of deep water.*

*Convection and deep water formation do not only depend on surface freshwater flux, but are more generally controlled by the surface buoyancy flux, which also depends on net heat losses and the temperature at the sea surface. The temperature dependence of the surface buoyancy flux arises from the nonlinear equation of state of seawater, and in particular from the temperature dependence of the thermal expansion coefficient* (Roquet et al., 2015). *Perturbations to the climate will affect both the net surface freshwater flux, as a result of changes in the hydrological cycle, and the surface temperature, with intricate implications for AMOC stability. The effect of climate on AMOC stability has been investigated in relatively few studies, mainly by changing the concentration of atmospheric $CO_2$* (Brown and Galbraith, 2016; Klockmann et al., 2018; Galbraith and de Lavergne, 2019). *These studies show a generally stronger AMOC in equilibrium with higher $CO_2$, but mostly focused on climates colder than present. Recently, Gérard and Crucifix (2024) have performed model simulations with slowly increasing and decreasing $CO_2$, producing an AMOC hysteresis in $CO_2$ space and suggesting an AMOC weakening in equilibrium with a warmer climate.*

*For an improved understanding of past and future AMOC evolution it is important to consider changes in climate and changes in the surface ocean freshwater balance due to changing land ice volume, since both play an important role for AMOC stability. Here we use an Earth system model to systematically explore the combined effect of surface FWF and climate on AMOC stability. The effect of external FWF is quantified by running experiments with FWF in different latitudinal belts in the North Atlantic, while the effect of climate is explored by varying the atmospheric $CO_2$ concentration, which is one of the main factors driving past and future climate changes.*

- I do not believe that the authors can claim to be the first to attempt to draw a landscape of AMOC stability in the CO2 x freshwater forcing space is true, see Brown and Galbraith 2016 for instance. However, I would say that this paper presents the most comprehensive and robust method up to date. If this claim only applies to modern-day studies of the AMOC, it needs to be clearly stated and put in context with palaeo studies.

To our knowledge we are the first who "have performed a systematic analysis of the AMOC stability in the FWF–CO2 space". Just to clarify the terminology: under stability (diagram) or phase portrait of the AMOC, we understand the 2D extension of the classical 1D Stommel's hysteresis diagram, which depicts the strength of different possible, stable AMOC modes and the positions of bifurcation transitions between different modes. All previous studies explore either one or the other dimension but not the two together. As far as the Brown and Galbraith (2016) paper is concerned, it does not aim to produce a stability diagram of AMOC. They performed a set of quasi-equilibrium simulations of the climate for different combinations of CO2, ice sheets and orbital parameters, as well as several transient water hosing experiments.

- Significant mode shifts and overshoots on the hysteresis cycles are not discussed in the text. In particular, in Figure 1, the transition in the red solid line around 0.05Sv is remarkable: Is it different from an overshoot? Why is it sustained for about 1000 years? Could this be an occurrence of millennial-scale variability? Could you link this to Willeit 2024?

We have added the following text to clarify this:
*The associated 'overshoot' in Fig. 1 is the result of a damped oscillation caused by the crossing of the bifurcation point between Modern and Weak AMOC states. In the presence of noise, such oscillations can become quasi-periodic, as shown in Willeit et al. (2024).*

- The definition of the different states comes too late in the paper and lacks precision. How do you define the different modes, using the AMOC index, the mixed layer depth or manually? Additionally, How do you calculate the AMOC index? What references did you use for the modern-day deep water formation sites, and how do they compare to your modern mode? Over what time slices was Figure 3 plotted?

The different modes are defined based on the mixed layer depth in different regions in the North Atlantic, as detailed below. This has been explicitly added to the caption of Fig. 4. The modern-day is characterized by deep water formation in the Labrador Sea and the Nordic Seas, which matches the locations with a deep mixed layer in the Modern AMOC state. Similarly to Fig. 6, also the AMOC states in Fig. 3 are for the boundary conditions of 400 ppm and 0.05 Sv, where all 4 AMOC states co-exist in the model. This has been added to the caption for clarification.

- The interpretation of the freshwater flux needs to be clarified in this article, and it becomes a problem when comparing pre-industrial to modern conditions. Would it not be more accurate to account for changes in CO2 and freshwater forcing when comparing the two? Otherwise, what is the point of using a two-dimensional landscape? In addition, the following statement from the conclusion "Our results indicate a generally stronger and deeper AMOC at equilibrium under warmer climate conditions. This is in contrast to the projected AMOC weakening response to anthropogenic global warming […]" is only valid if one considers the sole CO2 effect, but freshwater forcing is expected to increase with Greenland melt, which could take us into a region of the landscape where all four modes exist. I think the comparison between past, present and future states should include a discussion about the role of excess freshwater induced by ice sheet melting.

Our stability diagram shows the equilibrium states of the climate (atmosphere-ocean-sea ice-land) system with fixed present-day ice sheets. Past or future freshwater flux from the

surrounding ice sheets is considered as the external forcing and thus is the second dimension. We have added the following text to the 'AMOC hysteresis in freshwater space' section to clarify this point:

*The FWF, as used in this study, represents perturbations to the freshwater balance of the North Atlantic by factors external to the climate (atmosphere-ocean-sea ice-land) system, namely from changing land ice volume that is not accounted for in our simulations because we use prescribed present-day ice sheets.*

Given the strongly transient nature of ongoing global warming it does not make sense to compare pre-industrial with present-day conditions in the context of the AMOC stability landscape. As already mentioned in the paper, the transient and equilibrium AMOC response to CO2 are fundamentally different, with an increase in AMOC strength with increasing CO2 under equilibrium conditions, but a weakening if the CO2 increase is fast. The non-trivial relation between future projected AMOC evolution and the stability landscape will be investigated in a forthcoming paper. Also, to avoid possible confusion on the interpretation of the red dot in Fig. 4, we have removed it. It was originally simply intended to show the present-day CO2 concentrations, but it could be misinterpreted in terms of the present-day AMOC state.

During glacial terminations, the anomalous freshwater flux into the North Atlantic exceeded 0.1 Sv and had a profound impact on the AMOC, while for present-day conditions and even for the whole considered range of CO2 (up to 2xCO2), the net freshwater flux from Greenland is small (<< 0.1 Sv (Otosaka et al., 2023; Briner et al., 2020; Calov et al., 2018)) and has little effect on the AMOC. We have added this information to the revised manuscript. For higher CO2, the future freshwater flux from Greenland can be significant, but for high CO2, Greenland will melt completely in around thousand years, which is much shorter than the time needed to trace the stability diagram. This highlights again that the only sensible way to deal with net freshwater from ice sheets when tracing the AMOC stability diagram is to treat it as external forcing.

We made it clear that the statement '*Our results indicate a generally stronger and deeper AMOC at equilibrium under warmer climate conditions…*' is strictly valid only if meltwater input from Greenland is small enough, which is clearly the case according to different studies cited above.

- The details about the construction of the stability landscape is lacking precision and its validity cannot be evaluated. This all the more important as you highlighted the dependence of the direction of variation in Figure 1,2 and B1.

We have added detailed information on how the stability landscape is constructed in the caption of Fig. A1:

*Simulation pathways used to explore the stability of the four different AMOC states in the combined CO2 and freshwater space plotted on top of the AMOC stability landscape shown in Fig. 4. The stability of the Off AMOC state in (a) was explored with simulations starting from a large FWF of +0.5 Sv and then gradually decreasing the FWF until the AMOC recovers, for all levels of CO2. The stability of the Strong AMOC state in (d) was tracked in simulations starting from a large negative FWF of -0.5 Sv and then gradually increasing the FWF. For the investigation of the stability of the (b) Weak and (c) Modern AMOC states, the starting point were pre-industrial conditions, marked by the black dot. The black arrows indicate the primary path through the CO2 and FWF space, from which subsequent*

*experiments with varying FWF in different directions are initialized (green arrows). Since the Strong AMOC state is not stable for CO2 lower than 280 ppm for the FWF range shown in the figure, the stability of the Modern AMOC state in (c) for CO2 lower than pre-industrial is diagnosed from simulations initialized with a large negative FWF of -0.5 Sv, similarly to what done in (d) for the Strong AMOC state. The rate of change of the forcing in all the experiments is 0.02 Sv/kyr for FWF and 2 %/kyr for CO2.*

**Specific comments**

**Abstract**

- L8 - Can you briefly define the OFF and Modern AMOC states?

We have added the following brief description:
*'Apart from an Off AMOC state with no North Atlantic deep water formation and a Modern-like AMOC with deep water forming in the Labrador and Nordic Seas as observed at present,…'*

- L11-12 ("In general, the model shows an increase in equilibrium AMOC strength for higher CO2 levels.") - This does not reflect the actual nature of the work, which goes way beyond this sole aspect. This statement is valid for the standard CO2 experiment in Figure 2, but not consistent with Figure 4 (ex. in Figure 4a, an increase of CO2 can trigger a weak mode). The abstract needs not to focus only on the CO2 experiment but also on the stability landscape.

See response to first general comment above.

**Introduction**

- L28 ("There is no consensus as to whether the AMOC is in a monostable or a bistable regime under present climate conditions") - This needs a reference; I am aware of discussions on the potential weakening of the AMOC, less so about the current state of the AMOC.

We have added references to the review paper by Weijer et al. (2019) and to Liu et al. (2017), who suggested that the AMOC is bistable at present based on observational constraints.

**Results**

- L59 ("In particular, there is a range of FWF over which the AMOC has two stable states has two stable states") – Can you be precise about the range of FWF you are talking about? It also depends on your definition of stability, as I would argue that the dip around 0.05 Sv in Figure 1 is a sign of instability.

We have specified that we are talking about: '*roughly between 0.01 and 0.17-0.18 Sv*'. Yes, the dip around 0.05 Sv marks the transition between the Modern and Weak AMOC states, but

doesn't change the fact that in terms of hysteresis curve the AMOC has two stable states for any given FWF in the above-mentioned range.

- L61 - Does "preindustrial conditions" mean 0 Sv in this case? Also, according to the methods, this experiment has pre-industrial $CO_2$ concentrations but a modern-day ice sheet. I would be more careful about using "pre-idustrial" conditions, what about "initial state" instead?

We have specified that pre-industrial conditions refer to 280 ppm of $CO_2$. We consider pre-industrial and present-day ice sheets to be the same, which is a very reasonable assumption.

- L63 ("suggesting a prominent role of convective instability") - Could you show that this is a convective instability, showing deep water formation sites activity, for example?

The potential energy released by convective mixing as a function of time in the simulations shown in Fig. 1 in the paper is shown in Fig. 1 below, separately for the Nordic Seas and the Labrador Sea. The figure clearly shows the abrupt decrease of convective activity associated with the AMOC transitions, providing a clear indication of convective instability occurring in the model.

[Figure]

*Figure 1 Potential energy released by convection in the Nordic Seas (above) and the Labrador Sea (below) as a function of time for the FWF hysteresis simulations with increasing FWF shown in Fig. 1 in the paper.*

- L65 - "This is the result of a collapse of deepwater formation […] of observed past Dansgaard-Oeschger events." : Here again, the manuscript is missing a plot with the deep water sites dynamics to verify this statement, and a reference about convection in D-O records.

We are not aware of any direct reconstructions of convection locations during DO events. We have therefore remove the 'observed' in the sentence and added a reference to Willeit et al (2024), which includes a figure with the simulated mixed layer depth during DO Stadials and Interstadials.

- L67 – A definition of what the authors mean by Off, Weak, Modern-day, Strong is needed at this point of the paper.

In the revised paper we have introduced the Off, Weak and Modern AMOC states already when discussing the results of Fig. 1 in Section 2, while a strict definition is given later, after all the different states have been introduced.

- L82 ("which is possibly more relevant for the ongoing global warming") - I disagree, both CO2 and meltwater discharge are relevant to future climate changes.

This statement can indeed be misleading. In terms of AMOC stability diagram it is true that future increases in CO2 are more relevant than meltwater, which is what we have also stated in the paper:
*For CO2 doubling the net freshwater flux into the northern North Atlantic increases by ~0.07 Sv, a relatively large freshwater flux, which is approximately **an order of magnitude higher than the net freshwater flux from the Greenland ice sheet simulated under similar temperatures** (e.g. Calov et al. (2018) and Briner et al. (2020)), and would roughly correspond to the rate of freshwater input resulting from the Greenland ice sheet melting completely over a time period of ~1500 years.*

However, since the transient AMOC response to global warming over the next few centuries is fundamentally different from the equilibrium response, we have deleted this statement in the revised paper.

- L85-93 - I find the wording of this section confusing. Is the rate of CO2 increase in this paper slower than the "slow" increase in Gérard and Crucifix? How do you explain that you see a strengthening of the AMOC when Gérard and Crucifix 2024 saw a decrease? Could this simply mean that the AMOC response to CO2 is highly uncertain and model dependent?

Considering also the comment by Gérard and Crucifix on our pre-print (https://doi.org/10.5194/egusphere-2024-1482-CC1), we have rewritten this paragraph:
*Gérard and Crucifix (2024) recently analyzed the AMOC response to a slow CO2 increase and found a gradual AMOC weakening and eventual collapse at CO2 above ~1500 ppm. This is in contrast to our results, which show an increase in AMOC strength with increasing CO2, at least up to a CO2 concentration of 560 ppm. It should be noted that both in Gérard and Crucifix (2024) and in our study the CO2 increase is slow enough to track the equilibrium AMOC response.*

- L105 ("For CO2 above ~250 ppm, the convection pattern resembles the present-day state with deep water forming in the Labrador Sea and in the Nordic Seas") - Are you talking about Willeit et al. 2024 or this manuscript?

We are talking about this manuscript, which should be clear form the fact that later in the same sentence we are explicitly referring to Fig. 3c, which shows the mixed layer depth for the *modern* AMOC state.

- L156 ("If the climate would be in equilibrium with present-day CO2 concentrations of ~420 ppm, the model suggests that the Modern AMOC state would not be stable, but that the AMOC would rather be in the Strong state instead") – Back to the point about the interpretation of future freshwater forcing, would the accelerated melt of Greenland not move the system along the Modern AMOC conditions diagonal instead? Otherwise, this is a major caveat of the analysis that needs to be discussed (although, arguably, the last paragraph of the discussion introduces this idea of model dependency).

Yes, the melt of Greenland would move the system in diagonal direction, but because freshwater input from Greenland melt is relatively small for the considered levels of global warming, the system would still be quite close to the FWF=0 axis. Also, as discussed above, this paper is really about the AMOC stability diagram and its relation to transient future climate and AMOC change will be the subject of further investigations.

- **Figure 4** - Good Figure but missing information about the criteria used for the clustering of the AMOC states. Is it purely based on the AMOC index, or on the mixed layer depth?

We have added the following text in the figure caption to clarify this:
*The different states are formally defined based on a critical threshold ($mld_{max}^{crit}$=600 m) of the maximum mixed layer depth ($mld_{max}$) in three separate regions in the North Atlantic, namely (i) the Nordic Seas, (ii) the Labrador Sea and (iii) the Barents and Kara Seas and the Nansen basin.*
*Off : $mld_{max}$< $mld_{max}^{crit}$ in (i-iii);*
*Weak: $mld_{max}$> $mld_{max}^{crit}$ in (i) and $mld_{max}$< $mld_{max}^{crit}$ in (ii-iii);*
*Modern: $mld_{max}$> $mld_{max}^{crit}$ in (i-ii) and $mld_{max}$< $mld_{max}^{crit}$ in (iii);*
*Strong: $mld_{max}$> $mld_{max}^{crit}$ in (i-iii).*

- **Figure 5** - I think this is the most important Figure of the paper and it deserves to be bigger. Also, why are the Off mode treated differently than the different modes? It makes the Figure difficult to interpret at first reading. Could you use different colours or shading and include all four states in the stability landscape?

We agree that this is the most important figure in the paper and have consequently made it larger. The Off state is in some sense fundamentally different from any of the 'on' states and it can not easily be framed in the general trend of AMOC becoming weaker with decreasing CO2 and increasing FWF, which is what the color shading is intended to represent. Representing the Off state with the same color scheme would be problematic when e.g. considering the region of the landscape where the Strong and Off state are stable. What color would be associated to the coexistence of these two extreme states?
We have instead replace the crosses, which indicate the stability region of the Off state, with a white hatched area. White does therefore represent the Off state in the new figure, as shown in Fig. 2 below. This does hopefully make the figure easier to read and interpret. We have also added the following text to the figure caption:

*The filled white area indicates where only the Off AMOC state is stable, while the white hatched area shows the domain where the Off AMOC state and one or more of the three 'on' AMOC states coexist.*

[Figure]

*Figure 2 Revised Fig. 5.*

- L181 - This is a good section and a convincing way to perform this analysis. However, as you showed, the sea ice extent is driving the most significant temperature changes, so I think it would be good to conclude on the impact of less extensive Arctic sea ice in future climate on these results.

We have added the following sentence to clarify this point:
*While the temperature differences in Fig. 6 are representative of the impact of the different AMOC states on climate, they are strictly valid only for an atmospheric CO2 of 400 ppm and a freshwater forcing of 0.05 Sv and could differ if the transition between AMOC states occurs under different boundary conditions.*

**Discussion**

- L186 - Can you say a word about the Modern and Off AMOC modes.

We have added some more information on these two states as follows:
*Apart from an Off AMOC state with no North Atlantic deep water formation and a Modern-like AMOC state with deep water forming in the Labrador and Nordic Seas as observed at present,...*

**SI**

- L256 - The paths used in **Figure A1** need to be clearly explained in the Figure or the text, otherwise it is impossible to validate the protocol in Figure 4. Do they correspond to the black or the green arrows? Do they include the standard experiments?

We now explain in detail the paths in Fig. A1 in an expanded caption:
*Simulation pathways used to explore the stability of the four different AMOC states in the combined CO2 and freshwater space plotted on top of the AMOC stability landscape shown in Fig. 4. The stability of the Off AMOC state in (a) was explored with simulations starting from a large FWF of +0.5 Sv and then gradually decreasing the FWF until the AMOC recovers, for all levels of CO2. The stability of the Strong AMOC state in (d) was tracked in simulations starting from a large negative FWF of -0.5 Sv and then gradually increasing the FWF. For the investigation of the stability of the (b) Weak and (c) Modern AMOC states, the starting point were pre-industrial conditions, marked by the black dot. The black arrows indicate the primary path through the CO2 and FWF space, from which subsequent experiments with varying FWF in different directions are initialized (green arrows). Since the Strong AMOC state is not stable for CO2 lower than 280 ppm for the FWF range shown in the figure, the stability of the Modern AMOC state in (c) for CO2 lower than pre-industrial is diagnosed from simulations initialized with a large negative FWF of -0.5 Sv, similarly to what done in (d) for the Strong AMOC state. The rate of change of the forcing in all the experiments is 0.02 Sv/kyr for FWF and 2 %/kyr for CO2.*

- **Figure B2** does not have a caption.

We apologize for having overlooked the missing caption and have added the following caption to Fig. B2:
*Change in the net freshwater flux into the ocean as a function of global temperature change in transient historical and future simulations under the SSP2-4.5 scenario until the year 2300 CE for (a) the northern North Atlantic and Arctic (north of 50° N) and (b) the whole Atlantic ocean. The solid line is for CLIMBER-X results and the circles represent CMIP6 model results.*

**Technical corrections**

*Introduction*

- L17 - Do you have an example of a "societal" change?

We have removed 'societal' as we focus only on the climate impact.

- L19 - Add Bellomo 2021 for state-of-the-art climate model references.

We have added references to Manabe and Stouffer (1993), Bellomo et al. (2021), Weijer et al. (2020), Weaver et al. (2012).

*Results*

- L51 – "Willeit et al. (2020)" to "(Willeit et al., 2020)

Has been fixed, thanks.

- L69 ("most AMOC hysteresis experiments to date have been performed with FWF at lower latitudes (usually between 20N and 50N)") – Needs references.

We have added a reference to Rahmstorf et al. (2005), Hu et al. (2012) and van Westen and Dijkstra (2023).

- L71 ("In our hysteresis experiment") – Change to "In Figure 1", there are multiple hysteresis experiments in this paper.

We have changed this to 'In our freshwater hysteresis experiment in Fig. 1'.

- L80 ("give the wrong impression that the AMOC Off state is also stable") – Is it a wrong impression or relative to the rate of meltwater discharge? Can you give an order of magnitude of the expected meltwater discharge for future melting or during a D-O event/Heinrich event?

The idea of the hysteresis experiments is to find the equilibrium states of the AMOC. To do this, the rate of change of freshwater forcing should be small enough to actually track the equilibrium states. The rate of change in our standard experiments is low enough to have confidence that the produced stability diagram is accurate enough. Obviously, the rate of change of 0.2 Sv/kyr corresponding to the dashed lines in Fig. B1 is not low enough and therefore the hysteresis curve produced with such rate of changes is wrong. These are idealized experiments that have little to do with past or future rates of meltwater discharge.

- L83 – "In an experiment where" to "In Figure 2"?

Has been changed as suggested.

- L94 ("two discrete transition") - what do you mean by discrete? Abrupt?

We have replaced discrete with abrupt.

- **Figure 2** - It is difficult to distinguish the solid and dotted lines on this Figure. Maybe two columns?

We made the continuous lines in (a) slightly transparent, so that the dashed lines are visible on top.

- L110 ("but this has been shown to not be a requirement for the existence of millennial-scale variability") – Vague, can you say more?

We have expanded the discussion of this point as follows:
*A narrow window of CO2 concentrations exists for which both convection patterns are stable for the same CO2, but in Willeit et al. (2024) it was shown that this bistability is not a requirement for the existence of millennial-scale variability, for which it is sufficient that the system is close enough to the bifurcation point between Modern and Weak AMOC states.*

- **Figure 3** - The Figure needs to be bigger. Please add labels to the colour-bars. Does seasonal mean winter in this case?

We have made the figure bigger and added labels to the colour bars. What is meant with 'seasonal maximum' is the monthly mean maximum over the year. We have changed this to 'maximum monthly mean…of the year' in the revised paper.

- L130 - "net freshwater flux" to "net surface freshwater flux".

Has been changed.

- L133 ("for CO2 doubling") – I would say "for double the amount of CO2" because you are not doing a CO2 doubling experiment, which, as you highlighted, could have a different impact on the AMOC.

Good point, thanks. Has been changed as suggested.

- L139 - "in the model the net surface freshwater flux into the whole Atlantic Ocean shows the opposite trend" to "the net surface freshwater flux into the whole Atlantic Ocean shows the opposite trend in our model"?

Has been changed.

- L144 - remove "a stabilizing effect"?

Has been removed.

- L150 - "can be investigating by tracing their stability through" to "can be investigated by tracing the AMOC response in"

Has been changed.

- L169 - "to explore the pure effect of the different AMOC states on climate" to "to isolate the effect of the changes of AMOC states on the climate".

Has been changed.

- **Figure 6**: Can you use a different colour scheme for the temperature?

As suggested also by reviewer #3, we have changed the color scale to make the figure easier to read.

*Discussion*

- L193 - "explains" to "demonstrates"?

Has been changed to avoid repetition in the use of 'explains'.

- L202 ("anthropogenic global warming") - Add Bellomo 2021 to the references for state-of-the-art climate models.

We have added a citation to Bellomo 2021.

*SI*

- **Figure B1**: Hard to distinguish the solid and dotted lines on this Figure. Maybe two columns?

We think that it is useful to see all three curves in the same figure to have a direct comparison between the three rates and the different curves seem to be well distinguishable where they don't overlap, which is what we are interested in.

**References**

Bellomo, K., Angeloni, M., Corti, S., and von Hardenberg, J.: Future climate change shaped by inter-model differences in Atlantic meridional overturning circulation response, Nat. Commun., 12, 1–10, https://doi.org/10.1038/s41467-021-24015-w, 2021.

Briner, J. P., Cuzzone, J. K., Badgeley, J. A., Young, N. E., Steig, E. J., Morlighem, M., Schlegel, N. J., Hakim, G. J., Schaefer, J. M., Johnson, J. V., Lesnek, A. J., Thomas, E. K., Allan, E., Bennike, O., Cluett, A. A., Csatho, B., de Vernal, A., Downs, J., Larour, E., and Nowicki, S.: Rate of mass loss from the Greenland Ice Sheet will exceed Holocene values this century, Nature, 586, 70–74, https://doi.org/10.1038/s41586-020-2742-6, 2020.

Brown, N. and Galbraith, E. D.: Hosed vs. unhosed: Interruptions of the Atlantic Meridional Overturning Circulation in a global coupled model, with and without freshwater forcing, Clim. Past, 12, 1663–1679, https://doi.org/10.5194/cp-12-1663-2016, 2016.

Calov, R., Beyer, S., Greve, R., Beckmann, J., Willeit, M., Kleiner, T., Rückamp, M., Humbert, A., and Ganopolski, A.: Simulation of the future sea level contribution of Greenland with a new glacial system model, Cryosphere, 12, 3097–3121, https://doi.org/10.5194/tc-12-3097-2018, 2018.

Galbraith, E. and de Lavergne, C.: Response of a comprehensive climate model to a broad range of external forcings: relevance for deep ocean ventilation and the development of late Cenozoic ice ages, Clim. Dyn., 52, 653–679, https://doi.org/10.1007/s00382-018-4157-8, 2019.

Ganopolski, A. and Rahmstorf, S.: Rapid changes of glacial climate simulated in a coupled climate model., Nature, 409, 153–8, https://doi.org/10.1038/35051500, 2001.

Gérard, J. and Crucifix, M.: Diagnosing the causes of AMOC slowdown in a coupled model: a cautionary tale, Earth Syst. Dyn., 15, 293–306, https://doi.org/10.5194/esd-15-293-2024, 2024.

Hu, A., Meehl, G. A., Han, W., Timmermann, A., Otto-Bliesner, B., Liu, Z., Washington, W. M., Large, W., Abe-Ouchi, A., Kimoto, M., Lambeck, K., and Wu, B.: Role of the Bering Strait on the hysteresis of the ocean conveyor belt circulation and glacial climate stability, Proc. Natl. Acad. Sci. U. S. A., 109, 6417–6422, https://doi.org/10.1073/pnas.1116014109, 2012.

Klockmann, M., Mikolajewicz, U., and Marotzke, J.: Two AMOC states in response to decreasing greenhouse gas concentrations in the coupled climate model MPI-ESM, J. Clim., 31, 7969–7984, https://doi.org/10.1175/JCLI-D-17-0859.1, 2018.

Liu, W., Xie, S. P., Liu, Z., and Zhu, J.: Overlooked possibility of a collapsed atlantic meridional overturning circulation in warming climate, Sci. Adv., 3, 1–8, https://doi.org/10.1126/sciadv.1601666, 2017.

Manabe, S. and Stouffer, R. J.: Century-scale effects of increased atmospheric $CO_2$ on the ocean–atmosphere system, Nature, 364, 215–218, https://doi.org/10.1038/364215a0, 1993.

Otosaka, I. N., Shepherd, A., Ivins, E. R., Schlegel, N. J., Amory, C., Van Den Broeke, M. R., Horwath, M., Joughin, I., King, M. D., Krinner, G., Nowicki, S., Payne, A. J., Rignot, E., Scambos, T., Simon, K. M., Smith, B. E., Sørensen, L. S., Velicogna, I., Whitehouse, P. L., Geruo, A., Agosta, C., Ahlstrøm, A. P., Blazquez, A., Colgan, W., Engdahl, M. E., Fettweis, X., Forsberg, R., Gallée, H., Gardner, A., Gilbert, L., Gourmelen, N., Groh, A., Gunter, B. C., Harig, C., Helm, V., Khan, S. A., Kittel, C., Konrad, H., Langen, P. L., Lecavalier, B. S., Liang, C. C., Loomis, B. D., McMillan, M., Melini, D., Mernild, S. H., Mottram, R., Mouginot, J., Nilsson, J., Noël, B., Pattle, M. E., Peltier, W. R., Pie, N., Roca, M., Sasgen, I., Save, H. V., Seo, K. W., Scheuchl, B., Schrama, E. J. O., Schröder, L., Simonsen, S. B., Slater, T., Spada, G., Sutterley, T. C., Vishwakarma, B. D., Van Wessem, J. M., Wiese, D., Van Der Wal, W., and Wouters, B.: Mass balance of the Greenland and Antarctic ice sheets from 1992 to 2020, Earth Syst. Sci. Data, 15, 1597–1616, https://doi.org/10.5194/essd-15-1597-2023, 2023.

Rahmstorf, S., Crucifix, M., Ganopolski, A., Goosse, H., Kamenkovich, I., Knutti, R., Lohmann, G., Marsh, R., Mysak, L. A., Wang, Z., and Weaver, A. J.: Thermohaline circulation hysteresis: A model intercomparison, Geophys. Res. Lett., 32, L23605, https://doi.org/10.1029/2005GL023655, 2005.

Roquet, F., Madec, G., Brodeau, L., and Nycander, J.: Defining a simplified yet "'Realistic'" equation of state for seawater, J. Phys. Oceanogr., 45, 2564–2579, https://doi.org/10.1175/JPO-D-15-0080.1, 2015.

Smith, R. S. and Gregory, J. M.: A study of the sensitivity of ocean overturning circulation and climate to freshwater input in different regions of the North Atlantic, Geophys. Res. Lett., 36, 1–5, https://doi.org/10.1029/2009GL038607, 2009.

Weaver, A. J., Sedláček, J., Eby, M., Alexander, K., Crespin, E., Fichefet, T., Philippon-Berthier, G., Joos, F., Kawamiy, M., Matsumoto, K., Steinacher, M., Tachiiri, K., Tokos, K., Yoshimori, M., and Zickfeld, K.: Stability of the Atlantic meridional overturning circulation: A model intercomparison, Geophys. Res. Lett., 39, 1–8, https://doi.org/10.1029/2012GL053763, 2012.

Weijer, W., Cheng, W., Drijfhout, S. S., Fedorov, A. V., Hu, A., Jackson, L. C., Liu, W., McDonagh, E. L., Mecking, J. V., and Zhang, J.: Stability of the Atlantic Meridional Overturning Circulation: A Review and Synthesis, J. Geophys. Res. Ocean., 124, 5336–5375, https://doi.org/10.1029/2019JC015083, 2019.

Weijer, W., Cheng, W., Garuba, O. A., Hu, A., and Nadiga, B. T.: CMIP6 Models Predict Significant 21st Century Decline of the Atlantic Meridional Overturning Circulation, Geophys. Res. Lett., 47, https://doi.org/10.1029/2019GL086075, 2020.

van Westen, R. M. and Dijkstra, H. A.: Asymmetry of AMOC Hysteresis in a State-Of-The-Art Global Climate Model, Geophys. Res. Lett., 50, https://doi.org/10.1029/2023GL106088, 2023.

**Reviewer #2**

I thank the author team for their interesting contribution. The study finds an interesting AMOC stability landscape in CLIMBER-X by systematically varying freshwater forcing and atmospheric CO2. I have not seen such a comprehensive study of AMOC stability across varying backgrounds before, and I think it provides an intriguing template to move beyond single-perturbation AMOC studies and broaden our understanding of the processes affecting AMOC stability on multi-millennial time scales. This merits publication in itself, but I think the authors could increase the relevance of their manuscript by expanding their discussion of the implications of their findings for our understanding of past or future AMOC states. I recommend publication with minor corrections.

We would like to thank the Reviewer for the positive appraisal of our work and the comments on our manuscript.
Following the Reviewers' suggestions below, we have now expanded the discussion on the implications of our results for the interpretation of AMOC states, both for the past (i.e. Pliocene) and for the future.

**Main points**

1. I did not understand how many and which simulations were run in total to explore the CO2-FWF space. Was each combination of CO2 and FWF run for initial 'on' and 'off' states? Were the 'pathway' simulations run transiently, or iteratively run into equilibrium?

To clarify which simulations were performed in the CO2-FWF space, we have added more detailed information about the paths in Fig. A1 in an expanded caption:
*Simulation pathways used to explore the stability of the four different AMOC states in the combined CO2 and freshwater space plotted on top of the AMOC stability landscape shown in Fig. 4. The stability of the Off AMOC state in (a) was explored with simulations starting from a large FWF of +0.5 Sv and then gradually decreasing the FWF until the AMOC recovers, for all levels of CO2. The stability of the Strong AMOC state in (d) was tracked in simulations starting from a large negative FWF of -0.5 Sv and then gradually increasing the FWF. For the investigation of the stability of the (b) Weak and (c) Modern AMOC states, the starting point were pre-industrial conditions, marked by the black dot. The black arrows indicate the primary path through the CO2 and FWF space, from which subsequent experiments with varying FWF in different directions are initialized (green arrows). Since the Strong AMOC state is not stable for CO2 lower than 280 ppm for the FWF range shown in the figure, the stability of the Modern AMOC state in (c) for CO2 lower than pre-industrial is diagnosed from simulations initialized with a large negative FWF of -0.5 Sv, similarly to what done in (d) for the Strong AMOC state. The rate of change of the forcing in all the experiments is 0.02 Sv/kyr for FWF and 2 %/kyr for CO2.*

2. The existence of a 'strong' AMOC mode above 370 ppm is a key result of the study. The authors already compare their results to other model studies and mention that it shows the potential for stronger-than-present AMOC states in the past. Given that this CO2 threshold is close to both current and Pliocene CO2 concentrations, and that the Arctic might have been seasonally ice-free in the Eemian, it would be interesting to discuss this result also in the context of observations and proxy data. Is this solution

purely theoretical or is there evidence that such a circulation pattern has existed in the past?

There is indeed some proxy-based evidence from the Pliocene suggesting a stronger AMOC in the higher CO2 world of the Pliocene. We have added the following to the revised manuscript:

*Our results suggest that a different mode of AMOC (the Strong mode) was possible during past warm climate conditions. The Pliocene was the most recent period in Earth's history with elevated atmospheric CO2 concentrations of ~400 ppm (Martínez-Botí et al., 2015; Seki et al., 2010), which, according to our results, would be high enough to push the AMOC to a Strong state. There is indeed proxy-based evidence of a stronger-than-present AMOC in the Pliocene (Raymo et al., 1996; Ravelo and Andreasen, 2000) with an increased northward ocean heat transport in the Atlantic (Dowsett et al., 1992), which is consistent with sea surface temperature reconstructions for this period showing warmer conditions in the North Atlantic (McClymont et al., 2020). Climate models also tend to produce a stronger AMOC under mid-Pliocene conditions, although with considerable spread (Zhang et al., 2021; Weiffenbach et al., 2023).*

The Eemian is probably not very relevant in the context of our paper, as the climate change is induced mainly by a very different orbit at that time, and the effect of orbital configuration on AMOC is not considered in our paper.

3.  Besides the 'strong' AMOC state, the study shows that the 'off' state is also stable for high CO2 concentrations. If I understand correctly, this is the conclusion from simulations that were initialised with an 'off' state at high CO2. What forcing is required to transition from a 'strong' AMOC state into an 'off' state under high CO2?

Yes, the stability of the Off state is derived from simulations starting with an initial large FWF hosing of +0.5 Sv. This is now more explicitly stated in the caption of Fig A1, as explained in the response to comment 1 above. Actually, as shown in Figs. 4 and 5, already a freshwater flux of 0.2 Sv is sufficient to bring the AMOC to an Off state under any CO2 concentration.

4.  What forcing is required to tip from a 'modern' into a 'strong' AMOC? What are the climatic impacts of this shift? Do the authors think that tipping into an 'off' or 'strong' AMOC is possible under persistent anthropogenic climate change?

If we understand this correctly, here the Reviewer is concerned about transient future climate change. This can of course only partly be addressed with our equilibrium experiments presented in the paper. Nevertheless, it is clear from our results that both the Off and Strong AMOC states are possible outcomes of ongoing global warming, and whether one or the other state will eventually be reached depends probably largely on the rate and amplitude of future warming.

To at least partly address this question, we have added the following to the discussion:
*It is hence in principle possible that slightly different future global warming trajectories could lead in one case to an irreversible (on multi-centennial time scales) AMOC shutdown and in another case to a transient AMOC weakening followed by a transition into a Strong AMOC state, eventually resulting in fundamentally different climate conditions in the North Atlantic. Transient model simulations under future emission scenarios will have to be performed to explore this possibility.*

**Minor points**

Page 12: Please add an explanation of the grey and black arrows and the black dots to the figure caption or a legend.

We have expanded the description of this figure, as explained in the response to the main comment 1 above.

Page 14: Please add a caption for Fig B2.

We apologize for having overlooked the missing caption and have added the following caption to Fig. B2:
*Change in the net freshwater flux into the ocean as a function of global temperature change in transient historical and future simulations under the SSP2-4.5 scenario until the year 2300 CE for (a) the northern North Atlantic and Arctic (north of 50° N) and (b) the whole Atlantic ocean. The solid line is for CLIMBER-X results and the circles represent CMIP6 model results.*

**Reviewer #3**

Using a fast, low-resolution Earth system model, the authors present a systematic and comprehensive exploration of the dynamical stability landscape of the AMOC under quasi-equilibrium forcing conditions. Notably, AMOC stable states are explored in a combined phase space of atmospheric CO2 concentration and North Atlantic freshwater forcing. One of their main results is the possibility for a presence of more than two stable AMOC states if the present-day CO2 concentration and freshwater forcing are run into equilibrium. Among these, there is a stable "off" state, but also a stable "strong" state with deep water formation in the Kara Sea.

This study is a very timely and useful contribution to the discussion about the near-term evolution of the AMOC – a discussion that has very recently gained much traction again. Knowledge of AMOC stability in quasi-equilibrium climates – as opposed to the fast-changing transient that is our reality – is essential context for the scientific understanding of the latter.

The paper is very well written and has a clear layout. I offer a few comments for the authors to consider while recommending the ms. for publication without a further round of reviews.

We would like to thank the Reviewer for the positive appraisal of our work and the comments on our manuscript.

**Comments**

The study explores a range of $CO_2$ concentrations between 180 and 560 ppm. Of course this is a reasonable choice given the pre-industrial concentration. Sadly though, it is not entirely unlikely that the Earth System might have a $CO_2$ concentration larger that 560 ppm in the foreseeable future. Could the authors comment on why they didn't explore that part of the phase space?

We restricted our study to concentrations below 560 ppm for two main reasons:
1) there is nothing fundamentally different happening in the model for CO2 concentrations between 560 and ~800 ppm that would warrant its inclusion in the paper,
2) for CO2 above ~800 ppm the model produces pronounced spontaneous oscillations in the formation of Antarctic bottom water, which then also affects the AMOC and substantially complicates the interpretation of the results. This feature could potentially be addressed in a separate study.

CLIMBER-X does not have internal interannual climate variability, as stated on l.220. The presence of such variability could lead to some states being indistinguishable (as said on l.222). Another possibility is that a stable state, while technically present, is occupied only with a very small probability (Monahan [2002], JPO 32, 2072-2085).

Indeed, the presence of internal variability ("noise") can destabilise some modes of the AMOC circulation that are stable in the absence of noise. The presence of noise can lead to spontaneous transitions between different AMOC modes, as demonstrated in another paper (Willeit et al., 2024 Climate of the Past Discussion) devoted to the Dansgaard-Oeschger events. The presence of a strong internal variability in GCMs can make the tracing of the AMOC stability diagram like shown in Fig. 4 problematic, if not impossible. In this sense,

the absence of such variability in CLIMBER-X is actually an advantage of the model, since it allows us to obtain the phase portrait of the system without noise and then to simulate realistic dynamical behaviour of the system by adding the noise.
We have added this information to the last paragraph of the paper:
*Since CLIMBER-X does not produce internal interannual climate variability it is possible that different modes of the AMOC, which are distinct in our simulations, may not be distinguished in the presence of strong variability (e.g. Monahan, 2002). The presence of noise can also lead to spontaneous transitions between different AMOC modes, as demonstrated in Willeit et al. (2024). Intrinsic internal variability in general circulation models can make the tracing of the AMOC stability landscape problematic. In this sense, the absence of such variability in CLIMBER-X is actually an advantage of the model, since it allows to obtain the phase portrait of the system without noise and then to simulate realistic dynamical behaviour of the system by adding the noise.*

For the three quantities plotted in Figure 2 we need the spatial context. Where, in e.g. latitude and depth, did you diagnose the AMOC maximum and the AMOC heat transport maximum? Is the delta net FW in panel (c) diagnosed over the entire surface of the Atlantic basin?

The AMOC strength is defined as the maximum of the AMOC streamfunction deeper than 700 m. This has been explicitly added to the caption in Fig. 1 and 2. The 'AMOC heat transport' label in panel 2b should read 'Atlantic heat transport' and is further explained in the caption: 'maximum meridional heat transport by the ocean in the Atlantic' should be clear enough, as the meridional heat transport is a function only of latitude and the maximum therefore unambiguously refers to the latitude where the heat transport is largest.
Panel (c) shows the delta net FW separately for the whole Atlantic and the North Atlantic north of 50°N as stated in the legend. To make it clearer we have also added this information to the caption.

I have one concern that I'd like the authors to comment on. Figure 3g tells us that the overturning streamfunction of the "modern" AMOC state shows a complete absence of the abyssal AABW (Antarctic bottom water) cell. We know that in reality the AABW cell is present (Talley et al. [2003], J. Clim. 16, 3213-3226; Johnson [2008], DOI 10.1029/2007JC004477). Does that imply that the CLIMBER-X phase space significantly deviates from present-day conditions? Or should we say that the state labelled "weak" here is the actual modern-day state? Fig. 3f, after all, displays the correct shape of the streamfunction, and Fig. 2a suggests that the amount of overturning is realistic in this "weak" state too? If the model did have variability, perhaps it would convect in the correct regions too?

First, we would like to note that the term "Modern" does not mean "present-day". The AMOC states and mixed layer depth fields shown in Fig. 3 are all for the same boundary conditions of 400 ppm of $CO_2$ and FWF of +0.05 Sv, i.e. for the boundary conditions under which all four AMOC states exist in the model. The states shown in Fig. 3 are therefore fully consistent with the plots in Fig. 6. This was not clear in the original paper, but has been made explicit in the caption of the figure in the revised manuscript. For the present-day, the simulated vertical profile of the AMOC streamfunction at 26°N compares reasonably well with the observations from the RAPID array (Fig. 1 below). However, it is true that the simulated AMOC in the model reaches a bit too deep and fills the whole Atlantic. It is also noteworthy that most CMIP6 models show the opposite bias, with a too shallow simulated AMOC (see Fig. 1 below). At present, CLIMBER-X also simulates an AABW cell of a

reasonable strength (e.g. Fig. 11 in Willeit et al. (2022)).

In terms of the mixed layer depth it is clear that the Modern AMOC state resembles much more closely the observed present-day pattern with deep water forming in the Labrador and Nordic Seas (Fig. 3b,c) and at least the application of noise in the form of random perturbations in the surface freshwater flux in the northern North Atlantic does not affect the locations of deep water formation in the model.

We have added the Fig. 1 below to the methods section of the revised paper, together with the following text:

*In particular, the simulated present-day AMOC overturning profile at 26°N in the Atlantic is close to observations (Fig. A1), although it reaches a bit too deep. The present-day deep convection patterns compare well to ocean reanalysis in the North Atlantic (Fig. 13 in Willeit et al. (2022)).*

[Figure]

Figure 1 Profiles of the AMOC streamfunction at 26°N for the present-day. CLIMBER-X (black) is compared to observations from the RAPID array (blue) and a number of CMIP6 models (grey).

Could you consider a different type of colour scale for Figure 6? Currently it's really hard to work out the displayed colling patterns quantitatively using the small hue difference from one colour level to the next.

As suggested also by reviewer #1, we have changed the color scale to try to make the figure easier to read.

When discussing Figure 6 (around lines 170 to 180), could it be worthwhile mentioning that an AMOC transition as in Figure 2a, driven by increasing $CO_2$ concentration, would trigger a Modern to Strong transition, and thus a warming actually, with the negative of the patterns in Figure 6 b, f, j?

Absolutely, we have added the following text to the revised paper:
*A slow increase in the $CO_2$ concentration would trigger a Modern to Strong AMOC*

*transition as shown in Fig. 2a, and thus a warming actually, with the opposite sign of changes in Fig. 6b,f,j.*

I think for Figure B2 no legend was provided.

We apologize for having overlooked the missing caption and have added the following caption to Fig. B2:
*Change in the net freshwater flux into the ocean as a function of global temperature change in transient historical and future simulations under the SSP2-4.5 scenario until the year 2300 CE for (a) the northern North Atlantic and Arctic (north of 50° N) and (b) the whole Atlantic ocean. The solid line is for CLIMBER-X results and the circles represent CMIP6 model results.*

---

## Referee Report (RR1)

I thank the authors for their meticulous revisions, which resulted in an interesting and easily readable text. All my comments have been addressed and I recommend accepting the new manuscript version for publication.

---

## Referee Report (RR2)

Second round of review of Willeit and Ganopolski's "Generalized stability landscape of the Atlantic Meridional Overturning Circulation" by Yvan Romé, University of Leeds

**General comments**

Firstly, I would like to thank the authors for their fast and detailed response to the previous review. The authors addressed the comments in detail, and the manuscript was modified in depth to answer them.

The most recent version of the manuscript improves on the previous one and satisfactorily answers the main concerns raised during the previous round of reviews. In particular, the text is clearer, more precise, and more nuanced.

Regarding the previous comments:

- The authors' comment about the scope of the abstract is convincing, and the new version of the abstract is a good summary of the work conducted in this study.
- The introduction was significantly improved. It answered thoroughly my previous comments, and I find the new version succinct and impactful. I understand the authors' response regarding the freshwater forcing and millennial-scale literature. Shifting the narrative to the limitations of previous hysteresis studies is an elegant way to avoid going through the extensive references on freshwater hosing simulations, which is, I agree, not entirely relevant to the paper.
- The new version of the paper does an excellent job of highlighting the novelty of the FWF $x$ CO2 AMOC stability diagram. The distinction between the previous studies and this method is also more transparent.
- I appreciate that the authors discussed the potential occurrence of millennial-scale variability in Figure 1 around 0.05 Sv. I have additional comments about the interpretation of this hysteresis cycle, but they can be addressed as specific comments.
- The definition of modes is better defined in the manuscript and is rigorous. I believe, however, that this should be introduced at the same time as the modes in the manuscript. In Sections 2 and 3, and Figures 1 and 2, it sounds like the AMOC modes are only defined by the strength of the AMOC, which makes the identification of the modes vague when it is actually done precisely. This is particularly relevant in Figure 2 where the transition between the modern and strong AMOC modes is not evident on the AMOC index alone. As argued in my previous review, an additional Figure/panel with MLD time series in the different regions for

the hysteresis simulations could be a good way to illustrate this method. This could be in Supplementary information.

- The additional sentences about the discussion of the freshwater fluxes make the argument more precise and nuanced.
- The stages of the construction of the stability landscape construction are more apparent. However, I think this may still be missing a short justification for the chosen design. For instance, why map the OFF mode only by going from high FWF to low FWF, when both directions matter in a hysteresis cycle? This makes sense when referring back to Figures 1 and 2, but a couple of sentences about this would improve the fluidity of the reading.

Although I do not have any major modifications to suggest, there are three more general revision points that I would like to raise:

- I think the manuscript is missing a clear definition of what the authors mean by stability. This is particularly important in the interpretation of the hysteresis figures. For instance, line 74 (*The AMOC in the model is monostable under pre-industrial conditions"*): One could argue that this is not the case because both the overshoot and the modern AMOC branch exist at this point. Stability becomes more transparent in the slow FWF variation plot (0.005Sv/kyr), and it is not clear why the authors did not choose this experiment.
- I would also avoid the use of more or less stable, as it is not clear what this means (more or less potential modes accessible for a point in the phase space? Less likely transitions between multiple modes? Or even more or less stochastic variability"). For example in line 241 ("a $CO_2$ increase drives the AMOC towards a stronger and more stable AMOC"): visually, this is true in Figure 2, but Figure 5 indicates that a $CO_2$ increase from PI conditions means travelling from a monostable mode to two bistable modes, which, one could argue, is less stable. This confusion is also apparent in L219 ("Our AMOC stability landscape demonstrates why warm climates are generally stable and cold are unstable"): There again, different interpretations can arise from the reading of Figure 2 and Figure 5.
- Figure 5 is a great tool to map the theoretical states accessible at a point of the phase space. However, because these modes also depend on the system's history, they can contradict the result of the hysteresis experiments from Figures 1 and 2 and create confusion in the text. In addition to the two examples introduced in the previous point (L241 and L219), the argument in L182 ("an Off AMOC state can not be achieved by varying $CO_2$ alone, but only through a large enough FWF") is in contradiction with Figure 5. While these assertions are not wrong, they should be

presented with more caution and nuance to capture the complexity of the tools presented here.

The suggested revisions remain secondary, and I recommend publishing the manuscript after minor revisions without a new round of review.

**Specific comments**

**Abstract**
/

**Introduction**
/

**Results**

- Figure 1 – In the same way as the dip around 0.05 Sv was discussed, I believe the overshoot should also be mentioned in the text.
- L100 – I understand the authors' point in my previous review, but I am still unsure if *wrong* is the right word to use here. Otherwise, one could argue that the 0.02 Sv/kyr experiment also gives a wrong illustration of the hysteresis cycle, as it produces different transitions from the slow experiment (0.005 Sv/kyr).
- L115 ("The warmer the climate the further north do the sites of deepwater formation shift, following the northward retreat of sea ice") - I find this sentence in contradiction with some of the following points of this paper, which is not what this figure shows. "A stronger AMOC is generally associated with a northward shift of the sites of deepwater formation shift, following the northward retreat of sea ice"?

**Discussion**

- L214 ("for $CO_2$ below ~250 ppm or FWF in the latitudinal belt 50–70°N above ~0.05 Sv") and L215 ("for $CO_2$ above ~350 ppm or FWF in the latitudinal belt 50–70°N below ~-0.2 Sv") - I think this is oversimplifying the complex landscape presented in Figure 5. I would be more precise or remove these two assertions.
- L219 (". The fact that the AMOC is monostable under pre-industrial-like conditions explains why the AMOC always recovered at the end of glacial terminations, after

temporary shutdowns induced by the freshwater input (~0.1 Sv) from rapidly melting ice sheets") - I think this sentence needs more nuance: what do the authors mean by pre-industrial-like conditions, there needs to be a reference about $CO_2$ concentrations during the previous interglacial, and I do not think that this explains the recovery of the AMOC during glacial termination given the transient nature of this phenomenon. Nonetheless, this is indeed a convincing argument in favour of it, and the discussion about the Pliocene is a good addition to the paper.

**SI**

- /

**Technical corrections**

The technical corrections provided in the new version are satisfactory. I only have three minor points to raise:

- L51-53 – I am not sure I understand this sentence. Does this mean changes in $CO_2$ and changes in surface ocean freshwater balance due to modifications in climate and land ice volume? Please ignore if this is not what was implied.
- L103 (and Appendix A2) - It may be clearer to write 20ppm/kyr instead of 2%/kyr (0.002%/yr)? Or do is it increasing the $CO_2$ concentration by 0.002 % every year ($CO_2(n+1) = CO_2(n)*1.002$) ?
- L200 ("with a cooling of ~15∘C") - cooling by up to …

---

## Author Response (AR2)

Response to the second round of review of Willeit and Ganopolski's "Generalized stability landscape of the Atlantic Meridional Overturning Circulation" by Yvan Romé, University of Leeds.

We would like to thank the Reviewer again and provide the response to his comments in blue below.

**General comments**

Firstly, I would like to thank the authors for their fast and detailed response to the previous review. The authors addressed the comments in detail, and the manuscript was modified in depth to answer them. The most recent version of the manuscript improves on the previous one and satisfactorily answers the main concerns raised during the previous round of reviews. In particular, the text is clearer, more precise, and more nuanced.

Regarding the previous comments:

• The authors' comment about the scope of the abstract is convincing, and the new version of the abstract is a good summary of the work conducted in this study.

• The introduction was significantly improved. It answered thoroughly my previous comments, and I find the new version succinct and impactful. I understand the authors' response regarding the freshwater forcing and millennial-scale literature. Shifting the narrative to the limitations of previous hysteresis studies is an elegant way to avoid going through the extensive references on freshwater hosing simulations, which is, I agree, not entirely relevant to the paper.

• The new version of the paper does an excellent job of highlighting the novelty of the FWF x CO2 AMOC stability diagram. The distinction between the previous studies and this method is also more transparent.

• I appreciate that the authors discussed the potential occurrence of millennial-scale variability in Figure 1 around 0.05 Sv. I have additional comments about the interpretation of this hysteresis cycle, but they can be addressed as specific comments.

• The definition of modes is better defined in the manuscript and is rigorous. I believe, however, that this should be introduced at the same time as the modes in the manuscript. In Sections 2 and 3, and Figures 1 and 2, it sounds like the AMOC modes are only defined by the strength of the AMOC, which makes the identification of the modes vague when it is actually done precisely. This is particularly relevant in Figure 2 where the transition between the modern and strong AMOC modes is not evident on the AMOC index alone. As argued in my previous review, an additional Figure/panel with MLD time series in the different regions for the hysteresis simulations could be a good way to illustrate this method. This could be in Supplementary information.

We added two additional figures (see below) to the Appendix showing the maximum mixed layer depth in the three different regions used to categorize the four different AMOC states for two selected curves of the FW and CO2 hysteresis experiments shown in Fig. 1 and 2 in the paper. We refer to these two new figures in sections 2 and 3, when discussing the FW and CO2 freshwater hysteresis experiments and introducing the different AMOC states.

[Figure]

• The additional sentences about the discussion of the freshwater fluxes make the argument more precise and nuanced.

• The stages of the construction of the stability landscape construction are more apparent. However, I think this may still be missing a short justification for the chosen design. For instance, why map the OFF mode only by going from high FWF to low FWF, when both directions matter in a hysteresis cycle? This makes sense when referring back to Figures 1 and 2, but a couple of sentences about this would improve the fluidity of the reading.

We have added a few sentences to the main text to justify and clarify our approach:
*'The standard approach to tracing the AMOC stability diagram is to slowly change one of the control parameters (usually FWF) first in one direction and then in the opposite direction. However, such a method often fails to trace all equilibria, especially when more than two equilibrium states coexist at the same point in the phase space. Therefore, we combined the traditional approach, which works in our case for the Strong and Off modes, with a more sophisticated procedure where we alternate changes in FWF and CO2 space to trace the Modern and Weak states.'*

We have also added a few sentences to the Appendix:

*'The first step was to run experiments with increasing and decreasing FWF, starting from -0.5 Sv and +0.5 Sv, respectively, for all CO2 levels. These simulations are sufficient to trace the stability of the Off and Strong AMOC states (Fig. A2a,d), because (i) for large positive FWF the AMOC collapses under any CO2 concentration and (ii) for large negative FWF the AMOC always transitions to the Strong state. The stability analysis of the Modern and Weak AMOC states uses the pre-industrial state (CO2 of 280 ppm and zero FWF) as initial condition, but then requires a more sophisticated procedure to trace their stability through the 2D phase space (Fig. A2b,c).'*

Although I do not have any major modifications to suggest, there are three more general revision points that I would like to raise:

• I think the manuscript is missing a clear definition of what the authors mean by stability. This is particularly important in the interpretation of the hysteresis figures. For instance, line 74 ("The AMOC in the model is monostable under pre-industrial conditions"): One could argue that this is not the case because both the overshoot and the modern AMOC branch exist at this point. Stability becomes more transparent in the slow FWF variation plot (0.005Sv/kyr), and it is not clear why the authors did not choose this experiment.

In the paper we mostly use the term "AMOC stability" in a colloquial but traditional sense as a shortened form of "tracing the AMOC stability diagram in a phase space, i.e. studying the phase portrait of the system". When we use the term "stable equilibrium state", we understand "stability" in the normal mathematical sense, i.e. small perturbation does not cause large changes. But, of course, with a numerical model we can only find stable equilibrium states. So "stable equilibrium state" = "equilibrium state" in our case. However, because to track 'true' equilibrium one would need to change the boundary conditions infinitively slowly, what we are presenting is an approximation of these states. In this context, the overshoot is clearly a result of the slow but transient nature of our experiments. We added the following sentence to highlight this:
*'The overshoots are a result of the transient nature of our experiments and become less prominent with slower rates of FWF changes (Fig. B1).'*
We have chosen to show the experiments with a 0.02 Sv/kyr rate of FWF change to be consistent with the rate used to perform the experiments in the 2D CO2-FWF space, for which using the even slower rate of 0.005 Sv/kyr would be computationally too expensive.

• I would also avoid the use of more or less stable, as it is not clear what this means (more or less potential modes accessible for a point in the phase space? Less likely transitions between multiple modes? Or even more or less stochastic variability"). For example in line 241 ("a CO2 increase drives the AMOC towards a stronger and more stable AMOC"): visually, this is true in Figure 2, but Figure 5 indicates that a CO2 increase from PI conditions means travelling from a monostable mode to two bistable modes, which, one could argue, is less stable. This confusion is also apparent in L219 ("Our AMOC stability landscape demonstrates why warm climates are generally stable and cold are unstable"): There again, different interpretations can arise from the reading of Figure 2 and Figure 5.

This is a good point and generally we tried to be clear on this in the paper. However, in a few places we overlooked this and have fixed it now:
Line 241: *'CO2 increase drives the AMOC towards a stronger and more stable AMOC' -> 'CO2 increase drives the AMOC towards a stronger state'*
Line 219: *'Our AMOC stability landscape demonstrates why warm climates are generally stable and cold are unstable' -> 'Our AMOC stability landscape demonstrates that interglacial climates of the Quaternary are generally stable, because of the mono-stability of the AMOC under pre-industrial-like*

*conditions. The fact that the AMOC is monostable…. The existence of the Weak AMOC state has been shown by Willeit et al. (2024) to be related to Dansgaard-Oeschger events in the model, explaining the large AMOC variability observed during glacial times.'*

• Figure 5 is a great tool to map the theoretical states accessible at a point of the phase space. However, because these modes also depend on the system's history, they can contradict the result of the hysteresis experiments from Figures 1 and 2 and create confusion in the text. In addition to the two examples introduced in the previous point (L241 and L219), the argument in L182 ("an Off AMOC state can not be achieved by varying CO2 alone, but only through a large enough FWF") is in contradiction with Figure 5. While these assertions are not wrong, they should be presented with more caution and nuance to capture the complexity of the tools presented here.

The AMOC modes shown in Fig. 5 don't depend on the history. "History" is always related to time, whereas stability diagram is not. The problem we face is how to trace all possible AMOC equilibrium states in 2D space using a numerical model. As mentioned in the response to a comment above, the standard method does not work for all AMOC states, and for modern and weak states we have to use a rather tricky approach. But this is purely a technical issue, because in our case any existing stable state can be "accessed" from any initial point in phase space - one just has to find the right trajectory. Of course, it is still theoretically possible that there are some equilibria that we have not been able to access with our approach. It is true that Fig. 1 and 2 only capture some of the states shown in Fig. 5, but that is actually one of the reasons why we performed experiments in the 2D phase space. This is mentioned already in the text: '…*but our results indicate that the standard method of tracing hysteresis in the FWF space may not be enough to find all possible AMOC modes*.' Generally, in the real world we can only find equilibrium states by doing transient experiments. However, by using such a slow rate of change, we are confident that we are really close to the equilibrium states, at least much closer than any previous studies.
The argument in L182 is correct, in the sense that starting from any of the 'on' AMOC states it is not possible to reach the 'Off' state by just (slowly) changing CO2.

The suggested revisions remain secondary, and I recommend publishing the manuscript after minor revisions without a new round of review.

**Specific comments**

**Abstract**

/

**Introduction**

/

**Results**

• Figure 1 – In the same way as the dip around 0.05 Sv was discussed, I believe the overshoot should also be mentioned in the text.

We have added the following sentence to describe this:
*'When FWF is then slowly decreased again the AMOC recovers from the Off state with an overshoot at ~0.01 Sv, independently from the latitude at which the FWF is applied.'*

• L100 – I understand the authors' point in my previous review, but I am still unsure if wrong is the right word to use here. Otherwise, one could argue that the 0.02 Sv/kyr experiment also gives a wrong illustration of the hysteresis cycle, as it produces different transitions from the slow experiment (0.005 Sv/kyr).

We are here specifically referring to whether a given rate of change captures or not the AMOC stability under pre-industrial conditions. For that it is correct to say that the faster rate of change experiments do not reflect the true (quasi)equilibrium states. In general it is of course true that also a rate of 0.02 Sv/kyr can be too fast to capture the true equilibrium states accurately, but specifically for the pre-industrial conditions the default rate of change of 0.02 Sv/kyr is slow enough to capture the correct stability landscape (monostable), while the 10x faster rate of change is not.

• L115 ("The warmer the climate the further north do the sites of deepwater formation shift, following the northward retreat of sea ice") - I find this sentence in contradiction with some of the following points of this paper, which is not what this figure shows. "A stronger AMOC is generally associated with a northward shift of the sites of deepwater formation shift, following the northward retreat of sea ice"?

Has been reformulated as suggested.

**Discussion**

• L214 ("for CO2 below ~250 ppm or FWF in the latitudinal belt 50–70∘N above ~0.05 Sv") and L215 ("for CO2 above ~350 ppm or FWF in the latitudinal belt 50–70∘N below ~-0.2 Sv") - I think this is oversimplifying the complex landscape presented in Figure 5. I would be more precise or remove these two assertions.

We have removed these two assertions to keep the statement more general. The details can be seen in Fig. 5.

• L219 (". The fact that the AMOC is monostable under pre-industrial-like conditions explains why the AMOC always recovered at the end of glacial terminations, after temporary shutdowns induced by the freshwater input (~0.1 Sv) from rapidly melting ice sheets") - I think this sentence needs more nuance: what do the authors mean by pre-industrial-like conditions, there needs to be a reference about CO2 concentrations during the previous interglacial, and I do not think that this explains the recovery of the AMOC during glacial termination given the transient nature of this phenomenon. Nonetheless, this is indeed a convincing argument in favour of it, and the discussion about the Pliocene is a good addition to the paper.

We have modified the sentence to:
*'The fact that the AMOC is monostable for CO2 concentrations around 280 ppm, a typical value for interglacials, in the absence of FWF also explains why the AMOC always recovered at the end of glacial terminations, after temporary shutdowns induced by the freshwater input (~0.1 Sv) from rapidly melting ice sheets.'*
Interglacials typically last for ~5-10 kyrs, which is long time enough for the AMOC to equilibrate.

**SI**

/

**Technical corrections**

The technical corrections provided in the new version are satisfactory. I only have three minor points to raise:

• L51-53 – I am not sure I understand this sentence. Does this mean changes in CO2 and changes in surface ocean freshwater balance due to modifications in climate and land ice volume? Please ignore if this is not what was implied.

No, it means:
- changes in climate (which will affect temperature AND net surface freshwater flux)
- changes in surface freshwater flux due to land ice volume changes

• L103 (and Appendix A2) - It may be clearer to write 20ppm/kyr instead of 2%/kyr (0.002%/yr)? Or do is it increasing the CO2 concentration by 0.002 % every year (CO2(n+1) = CO2(n)*1.002) ?

It is actually an exponential CO2 increasing rate so that we get a roughly linear global temperature increase with time. The rate of change is 0.002%/yr, corresponding to CO2(n+1) = CO2(n)*1.00002, with a time step of 1 year. We have added the following to the Appendix A2 in order to further clarify this:
*'We have chosen an exponential \chem{CO_2} change rate in order to get a roughly linear global temperature response with time, considering the logarithmic dependence of the \chem{CO_2} radiative forcing.'*

• L200 ("with a cooling of ~15◦C") - cooling by up to …

Has been fixed.